# Spread spectrum SERS allows label-free detection of attomolar neurotransmitters

Wonkyoung Lee[1,2,3], Byoung-Hoon Kang[1,2], Hyunwoo Yang[1,2], Moonseong Park[1], Ji Hyun Kwak[1], Taerin Chung[1], Yong Jeong [1,2], Bong Kyu Kim[3] & Ki-Hun Jeong [1,2✉]

The quantitative label-free detection of neurotransmitters provides critical clues in understanding neurological functions or disorders. However, the identification of neurotransmitters remains challenging for surface-enhanced Raman spectroscopy (SERS) due to the presence of noise. Here, we report spread spectrum SERS (ss-SERS) detection for the rapid quantification of neurotransmitters at the attomolar level by encoding excited light and decoding SERS signals with peak autocorrelation and near-zero cross-correlation. Compared to conventional SERS measurements, the experimental result of ss-SERS shows an exceptional improvement in the signal-to-noise ratio of more than three orders of magnitude, thus achieving a high temporal resolution of over one hundred times. The ss-SERS measurement further allows the attomolar SERS detection of dopamine, serotonin, acetylcholine, γ-aminobutyric acid, and glutamate without Raman reporters. This approach opens up opportunities not only for investigating the early diagnostics of neurological disorders or highly sensitive biomedical SERS applications but also for developing low-cost spectroscopic biosensing applications.

[1] Department of Bio and Brain Engineering, Korea Advanced Institute of Science and Technology (KAIST), 291 Daehak-ro, Yuseong-gu, Daejeon 305-701, Republic of Korea. [2] KAIST Institute for Health Science and Technology (KIHST), KAIST, 291 Daehak-ro, Yuseong-gu, Daejeon 305-701, Republic of Korea. [3] Network Research Division, Electronics and Telecommunications Research Institute (ETRI), 218 Gajeong-ro, Yuseong-gu, Daejeon 350-700, Republic of Korea. ✉email: kjeong@kaist.ac.kr

Neurons in the brain release neurotransmitters at synapses to convey information to neighboring neurons. The precise amount of secreted neurotransmitters plays an important role in proper brain function; a small alteration of the neurotransmitter amount indicates some brain disorders[1–4]; thus, the concentrations of various neurotransmitters serve as candidate biomarkers for the early detection, prognosis, and real-time follow-up of neurological diseases or disorders. In particular, major neurotransmitters such as dopamine, serotonin, acetylcholine, γ-aminobutyric acid (GABA), and glutamate are highly associated with Alzheimer's disease (AD)[5–7], Parkinson's disease (PD)[8,9], schizophrenia[10], social anxiety, attention-deficit hyperactivity disorder (ADHD)[11], and Huntington's disease[12] in the central nervous system[13,14]. For instance, the level of acetylcholine involved in attentional processes and modulating excitatory neurotransmitters substantially decreases in AD[15]. This finding rationalizes the use of acetylcholinesterase inhibitors for AD treatment and supports the use of the neurotransmitter level in cerebrospinal fluid (CSF) as a biomarker[16]. Other changes in neurotransmitters such as dopamine for PD and schizophrenia and serotonin for depression are also well documented[8,17]. As a result, highly sensitive quantification is crucial for the quantitative in vitro bioassays of assorted neurotransmitters in serum or CSF.

The conventional separation and quantification of neurotransmitters in microdialysis samples often utilize high-performance liquid chromatography (HPLC)[18] coupled to either electrochemical or fluorescence detection[19] and a more recently developed technique, capillary electrophoresis laser-induced fluorescence detection[20]. An alternative method such as gas chromatography[21] or mass spectrometry[22] is often utilized for the quantification of drugs and other analytes. However, these methods still face some technical limitations for clinical uses due to tedious sample preparation, low detection sensitivity, and poor sample stability[23]. For example, a microdialysis sample of glutamate in the brain typically shows approximately several micromoles in basal concentration[24]. Moreover, the calibration complexity and the signal uncertainty still hinder the measurement of lower basal concentrations. Conventional separation techniques such as HPLC also require over a few tens of minutes to collect sufficient sample volume[25]. For instance, excitatory glutamatergic currents decay rapidly by hundreds of milliseconds due to diffusion, re-uptake, binding to receptors, or enzymatic breakdown. Consequently, the molecular identification of neurotransmitters still requires real-time and ultrasensitive detection methods. Surface-enhanced Raman spectroscopy (SERS) can serve as an effective alternative for the label-free quantification of various neurotransmitters[26,27]. However, the direct quantification of neurotransmitters in biological fluids by SERS remains a great challenge because of the extremely low concentration ($<10^{-10}$ M) of neurotransmitters in the cerebral extracellular fluid of neurological disease patients and the relatively small Raman cross-section of molecules[28–30]. In addition, recent techniques for the ultrasensitive monitoring of changes in neurological functions are required to improve the limit of detection (LOD)[27–31].

The SNR enhancement of SERS for the highly sensitive detection of neurotransmitters at extremely low concentrations can be achieved by using mainly plasmonic enhancement or signal processing. Plasmonic enhancement results from strong localized electromagnetic near fields observed in metal nanostructures in the visible range, originating from surface plasmons[32]. Extremely strong plasmonic hot spots at the nanogaps directly boost the major Raman signals[33]. The distinct shapes of the metal nanostructures from a highly anisotropic nanorod to polygonal nanostructures with a sharp tip produce enormous plasmonic fields, allowing detection down to the single-molecule level[34–36]. However, plasmonic SERS-active substrates mostly require high-cost

top-down nanofabrication methods or highly ordered nanoparticle synthesis to obtain stable and reproducible SERS signals for practical biochemical applications. In contrast, signal processing methods such as mathematical methods[37,38], time-resolved gating[39], and wavelength modulation[40], can efficiently reduce the background noise from SERS signals. For instance, mathematical methods including polynomial fitting[41], wavelet transformation[42], derivative processing[37], or principal component analysis (PCA)[38] eradicate fluorescence background signals without additional experimental setup or sample preparation. However, most previous methods are also still not suitable for either removing superimposed noise on a broad background from Raman spectra or increasing the sensitivity of SERS. For example, PCA causes potential artifacts in which unknown species distort signals and background drift results in uncertainties during the measurement, which may distract the validity of the acquired biochemical information. Both the time-resolved gating and wavelength modulation based on a short lifetime of Raman scattering also hamper practical biomedical applications due to the system cost as well as a relatively low SNR enhancement[37,38]. As a result, there exists a strong motivation for developing a novel approach for exceptional SNR enhancement in SERS detection, which enables label-free bioassays at the single-molecule level and quantitative diagnostics of neurotransmitters.

Here, we report the spread spectrum SERS (ss-SERS) detection of unlabeled neurotransmitters down to the attomolar-level concentration using spreading coded light excitation. The ss-SERS completely removes all the background noise, including fluorescence by using peak autocorrelation and near-zero cross-correlation of spreading codes. The experimental results of ss-SERS show a significant SNR enhancement of SERS signals and label-free detection of dopamine, serotonin, acetylcholine, GABA, and glutamate at attomolar concentrations.

## Results

**Working principle of ss-SERS.** The ss-SERS is based on the spread-spectrum technique, which is well known for attaining a high SNR and dynamic range in network applications, such as radio detection and ranging (radar), code-division multiple access (CDMA), or optical time domain reflectometer (OTDR)[43,44]. A conceptual description of ss-SERS is shown in Fig. 1a. Unlike continuous light excitation, a code sequence of pulsed light based on a spreading code excites target molecules on a SERS substrate, and then the SERS signals are encoded with the same pattern as the spreading code. A detector receives all the coded signals, including the SERS, autofluorescence, and background noise signals, resulting from the laser power and wavelength instability, shot noise, and thermal noise. The SERS signals are extraordinarily enhanced by using the peak autocorrelation between the spreading code and the coded SERS signals in the decoding process. Concurrently, all the background noise irrelevant to the spreading code or fluorescence signals encoded in a distorted pattern are completely filtered out due to the near-zero cross-correlation. The suppression of noise in SERS signals results in exceptional SNR enhancement. The ss-SERS detection method is consequently achieved by encoding the excitation light with a spreading code and decoding the SERS signals with peak autocorrelation and near-zero cross-correlation. The encoding and decoding principles for restoration and SNR enhancement of the noise-suppressed SERS signals are illustrated in Fig. 1b. Coded SERS signals are generated from molecular vibrations that are excited by using light modulation based on a spreading code of pseudorandom noise (PN) during the encoding process (the left side of Fig. 1b). The coded light and coded SERS signals are displayed as $S_{CW} * PN$ and $S_{SERS} * PN$, respectively, meaning

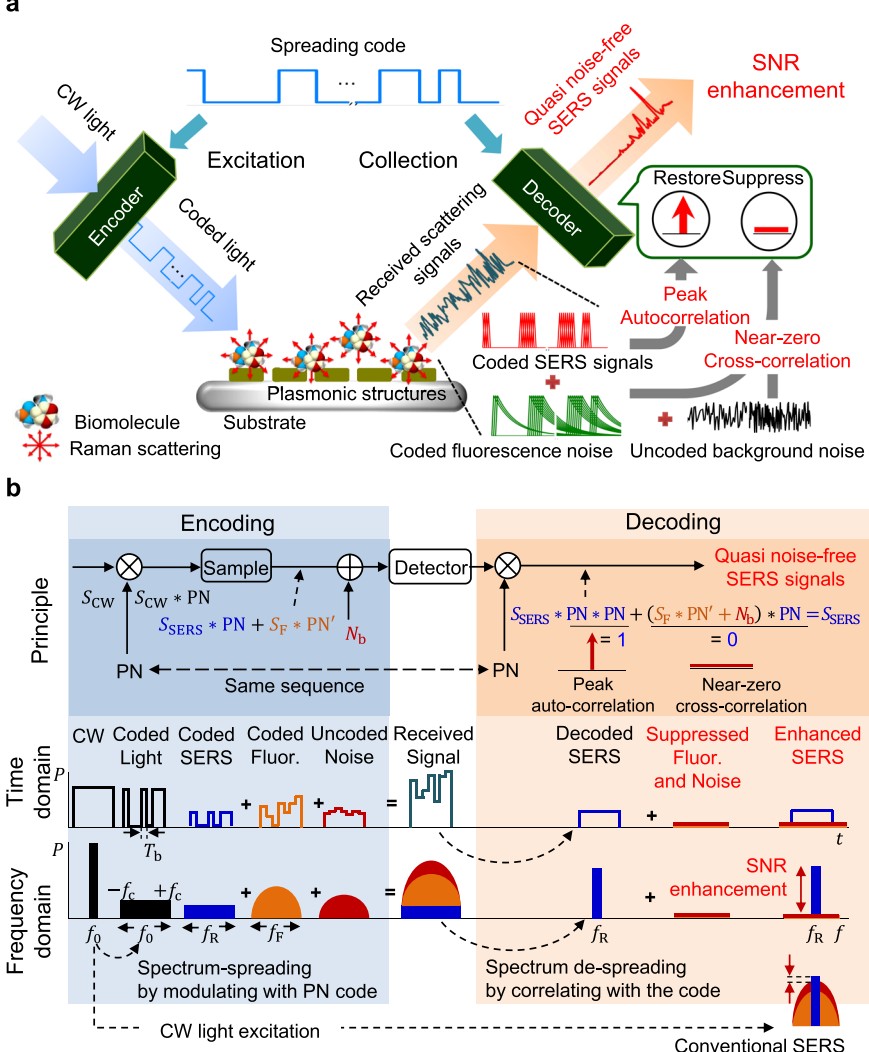

**Fig. 1 Conceptual description of spread spectrum SERS (ss-SERS) detection with spreading coded light excitation. a** Schematic illustration of ss-SERS detection with coded light excitation. A laser beam is encoded by using a spreading code of pseudorandom noise (PN) and is injected on a SERS substrate with Raman active molecules, where the SERS signals are simultaneously encoded with the coded excitation of light. Noise-suppressed SERS signals are reconstructed at the decoder by using peak autocorrelation between the spreading code and the coded SERS signals, whereas all noise, including fluorescence and background noise, is effectively eliminated due to a near-zero cross-correlation between the spreading code and the noise. **b** Encoding and decoding principles for the removal of background noise and the restoration of noise-suppressed SERS signals. $S_{CW}$, $S_{SERS}$, $S_F$, and $N_b$ represent the CW light excitation, original SERS signals, original fluorescence signals, and background noise signals, respectively. PN is the PN code used in the encoding process and PN′ is a distorted pattern from PN. The bit duration ($T_b$) and the modulation frequency ($f_c$) are inversely proportional. $P$, $t$, and $f$ represent the signal power, time, and frequency, respectively. In the time domain, the coded light and the coded SERS signals exhibit pulse sequences of the same pattern as the PN code of the encoding process, whereas the fluorescence signals are encoded in a distorted pattern from the PN code due to the lifetime of the fluorescence signals being longer than the bit duration. Consequently, the coded fluorescence and background noise are completely filtered out by correlating with the PN code of the encoding process due to the spectrum-spreading property of the spreading code, resulting in an exceptional SNR enhancement.

continuous wave (CW) light ($S_{CW}$) and original SERS signal ($S_{SERS}$) convoluted with the PN code (PN). The fluorescence signal ($S_F$) is encoded in a distorted pattern (PN′) from the spreading code because the lifetime of the fluorescence signal is longer than the bit duration at the modulation frequency; the coded fluorescence signal is represented as $S_F * PN′$. The background noise ($N_b$) is added to both the coded SERS signals and the coded fluorescence signals, and then all the signals are delivered to the detector ($S_{SERS} * PN + S_F * PN′ + N_b$). In the time domain, the coded light and coded SERS signals are represented as the same pattern as the PN code during the encoding process. Due to the spectrum spreading properties of the PN code, coded light, and coded SERS signals, the coded fluorescence

signals are spread out in the frequency domain, where all the noise is added (the bottom of Fig. 1b). The decoding process, i.e., spectrum de-spreading, is a series of operations that reduces the noise and restores the quasi-noise-free SERS signals. The original SERS signals are exactly reconstructed during the decoding process by using the peak autocorrelation of the spreading code and the coded SERS signals. The reconstruction process is mathematically expressed as $S_{SERS} * PN * PN = S_{SERS}$ because $PN * PN = 1$ is attributed to the peak autocorrelation of the spreading code. All noise, including the fluorescence signals, is effectively eliminated due to near-zero cross-correlation with the PN code (($S_F * PN′ + N_b) * PN = -1/N \approx 0$), where $N$ is the code length. Unlike the encoding process, only noise is spread out during the

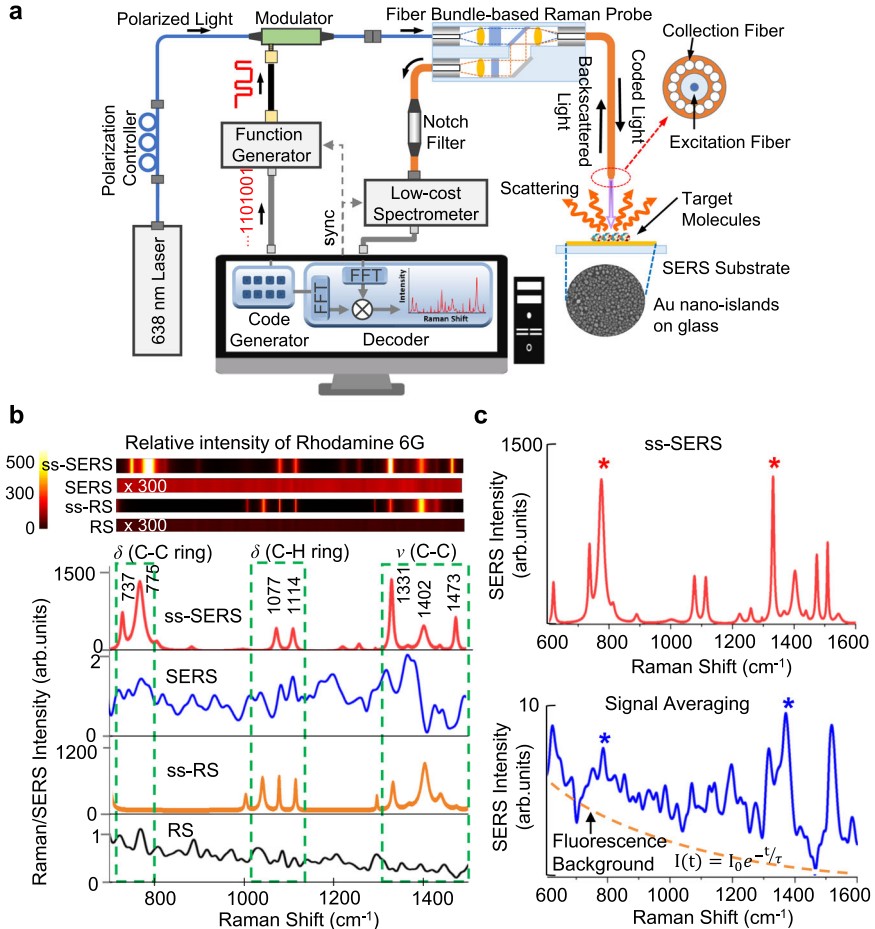

**Fig. 2 More than a two order of magnitude increased SNR of the spread spectrum SERS (ss-SERS). a** Experimental setup for the SERS measurement of ss-SERS consisting of an excitation light encoder for laser modulation, a signal decoder for deconvolution and a conventional SERS system for light excitation and SERS detection of target molecules. In the excitation light encoder, coded light is generated by modulating a laser beam with a PN code using an intensity modulator. A fiber bundle-based Raman probe launches the coded sequences into the target molecules on the SERS substrates and then receives the mixed signals with the coded SERS signals, coded Rayleigh scattering signals, coded fluorescence signals, and various system noise. The signal decoder restores quasi-noise-free SERS signals by correlating the detected signals with the identical PN code. **b** Raman or SERS spectra for Rhodamine 6G (R6G) at a 5 mM concentration measured by Raman spectroscopic equipment (RS), conventional SERS, ss-RS, and ss-SERS. The upper side shows the color maps of the measured Raman or SERS intensities. The ss-SERS peak intensity of R6G at 1331 cm$^{-1}$ is increased over 800 times that of SERS, and the ss-RS peak intensity at 1402 cm$^{-1}$ is increased over 1000 times that of RS. (The output power of the laser: 25 mW, the power at the sample: 1 mW, accumulation time: 10 s.) $\delta$ bending, $\nu$ stretching. **c** Comparison of the SERS spectra for ss-SERS and signal averaging. The ss-SERS peak intensity of R6G at 1331 cm$^{-1}$ is increased by over 150 times compared to the averaged SERS signals for the same measurement time. The SNR of the ss-SERS signals (5 mM rhodamine 6G, code length: 512 bits, measurement time: 10 s) is increased over two orders-of-magnitude compared to that of averaged SERS signals for the same measurement time.

decoding process, resulting in a significant enhancement of the SNR in the SERS signals. The correlation properties of the spreading code are directly related to the randomness and orthogonality of the spreading code. The spreading codes with excellent autocorrelation and cross-correlation properties, i.e., peak autocorrelation and near-zero cross-correlation, dramatically improve the SNR and effectively eliminate noise. Consequently, the SERS signals are clearly separated from all the background noise, reducing the distortion caused by the noise and resulting in a strong resistance to noise.

**Experimental demonstration of ss-SERS detection**. The ss-SERS method was experimentally demonstrated by adding an excitation light encoder for laser modulation and a signal decoder for deconvolution to a conventional SERS system (Fig. 2a). The light excitation is driven by a code generator, which produces a single continuous code sequence and the same code sequence is

repeatedly sent for a total measurement time. In the excitation light encoder, coded light is generated by modulating a laser beam (Ocean Optics LASER-638-LAB-FCA, center wavelength: 638 nm, output power: 25 mW) with a PN code using a LiNbO$_3$ intensity modulator (Jenoptik AM635, center wavelength: 635 nm, spectral bandwidth: ±20 nm, extinction: 500:1). A PN code consists of a pseudorandom binary sequence of bits, where each bit has a bit duration, i.e., the reciprocal of the modulation frequency. A fiber bundle-based Raman probe (Ocean Optics RIP-RPB-638-FC, excitation wavelength: 638 nm), consisting of excitation and collection fibers with a built-in laser line filter, launches the coded sequences into the target molecules on a nanoplasmonic SERS substrate. The Raman probe receives the coded SERS signals, coded Rayleigh-scattering signals, coded fluorescence signals, and various background noise. The Rayleigh-scattering signals are removed by the built-in laser line filter and then all other signals are transmitted into a low-cost spectrometer (Hamamatsu C10083CAH, spectral resolution:

1 nm, spectral range: 320–1000 nm, slit width: 10 μm). The decoding process restores quasi-noise-free SERS signals by correlating the detected signals with the identical PN code (Supplementary Fig. 1). The decoded SERS peaks of the target molecules are clearly obtained by subtracting the ss-SERS signals of a substrate material from the restored SERS signals (Supplementary Fig. 2). To extract the linewidth of the SERS peaks indicating the crystallinity of the material or the influence of various environments, the ss-SERS spectra of the Lorentzian line shape are finally obtained by substituting the intensities, center positions, and standard deviations of the calibrated decoded SERS peaks into the Lorentzian line shape function (Supplementary Fig. 3)[45]. In the experiment, 5 nm-thick Au nanoislands on a quartz glass substrate were selected as a SERS substrate (see "Methods" section for SERS substrate preparation). The Au nanoislands were simply fabricated by using the low-temperature solid-state dewetting of a thin film on a quartz glass wafer (Supplementary Fig. 4)[46]. The SNR enhancement of ss-SERS was confirmed by the SERS spectra for a reference molecule, rhodamine 6 G, at a concentration of 5 mM (see "Methods" section for sample preparation). The ss-SERS and spread spectrum Raman spectroscopy (ss-RS) of R6G, described in Fig. 2b, are compared with those of conventional SERS (SERS) and Raman spectroscopy (RS). Both ss-SERS and SERS used Au nanoislands on quartz glass as SERS substrates, whereas ss-RS and RS used quartz glass. The ss-SERS peak intensity of R6G at 1331 cm$^{-1}$ corresponding to the aromatic C–C stretching vibration of the xanthene ring and two NHC$_2$H$_5$ groups is observed to be more than 800 times greater than that of the SERS. It was further verified that the spreading coded light excitation is applicable not only to SERS but also to RS by measuring the SNR enhancement of the Raman signals of R6G at 5 mM. The intensity of the ss-RS peak at 1402 cm$^{-1}$ is observed to be more than 1000 times greater than that of the RS peak. The vibrational bands assigned to the measured ss-SERS peaks of R6G are presented in SI (Supplementary Table 1). The top of Fig. 2b shows a comparison of the color maps of the Raman or SERS intensities measured by RS, SERS, ss-RS, and ss-SERS. The major peaks of ss-RS and ss-SERS are clearly distinguishable due to the high SNR, unlike those of RS and SERS. In addition, the SNRs of the ss-SERS signals and the signal averaging signals were further compared in Fig. 2c. The SERS signals were averaged every 100 ms for 10 s. In the experiment, the ss-SERS signals for a code length of 512 bits are over two orders of magnitude greater than the averaged SERS signals for the same measurement time (Fig. 2c). As a result, the spreading codes with peak autocorrelation and near-zero cross-correlation substantially eliminate not only the background noise but also the fluorescence signals, unlike signal averaging, and further dramatically improve the SNR of the SERS signals.

**Quasi-noise-free SERS signals with high SNR**. The peak autocorrelation and near-zero cross-correlation of spreading codes play an important role in achieving quasi-noise-free SERS signals with a high SNR. Both the correlation properties are mainly determined by the code length and the modulation frequency. A single spreading code with high orthogonality consists of a balanced number of zero and one bits, where the bit duration is $T_b$ and the code length is $N$ (Fig. 3a). The spreading code has a delta-peak-like autocorrelation, which has a peak value of one at the zero time shift between the coded signals of the excited light and SERS as well as the sidelobe of $-1/N$ at all other time shifts within a code sequence. The time domain autocorrelation also represents the power spectral density (PSD) in the frequency domain after the Fourier transform, where the envelope shape is a sinc function, the main-lobe peak is proportional to $(N + 1)/N^2$,

and the spectral spacing is $1/(N \cdot T_b)$. Note that a single ideal orthogonal code of a random binary sequence has no autocorrelation sidelobe. The anti-noise performance factors mainly include the autocorrelation sidelobe, the main-lobe peak, and the spectral spacing. All the factors were numerically calculated by changing the code length (Fig. 3b). A PN code sequence of high orthogonality was generated by using the linear feedback shift register (LFSR) with maximum cycle length based on an irreducible primitive polynomial (Supplementary Fig. 5). Supplementary Fig. 5a lists the primitive polynomials of degrees from 2 to 20. The degree $n$ of the primitive polynomial is strongly associated with the code length, which is described as $2^n$ by adding a zero-stuff bit to compensate for the number imbalance between zero and one bit. For instance, a PN code sequence of 512 bits ($n = 9$) is created from the primitive polynomial of $x^9 + x^4 + 1$. In this calculation, the bit duration for high orthogonal codes was fixed at 4 ns corresponding to a modulation frequency of 250 MHz. The time decay of a single unit peak intensity of the SERS signal ($S_{SERS}$) was considered an exponential function with a Raman lifetime of 10 ps. The autocorrelation sidelobe depending on the code length was obtained from the maximum amplitude of the sidelobes. The calculated results clearly show that the main-lobe peak and the autocorrelation sidelobe approach the theoretical curve as the code length increases. As a result, the autocorrelation for the spreading code exhibits a delta-peak-like function and the power spectral density also shows equally low intensity at different frequencies; thus, both properties result in exceptional noise suppression as the code length increases.

The modulation frequency is also crucial for attaining the near-zero cross-correlation between the spreading code and the background noise signals, particularly from fluorescence. The cross-correlation between a spreading code sequence and all the noise, including the coded fluorescence signal and the background noise, were numerically calculated from the maximum amplitude of the cross-correlation function for modulation frequencies from 250 kHz to 250 MHz, where the code length was fixed at 512 bits (the left side of Fig. 3c). The time decay of a unit peak intensity of the fluorescence signal was considered to be an exponential function with a 10 ns fluorescence lifetime. The background noise was considered a white Gaussian random function with a mean of zero and variance of one. As the modulation frequency increases, the cross-correlation for the fluorescence decreases; however, the cross-correlation for the background noise remains near zero. As a result, both the fluorescence and the background noise are effectively suppressed at the same time by correlating with a spreading code at high modulation frequencies without requiring a separate bandpass filter or time-gated detector. The SNR of the ss-SERS major peak of 1331 cm$^{-1}$ for 5 mM R6G molecules was also measured at modulation frequencies of 250 kHz, 100, and 250 MHz (the right side on Fig. 3c). The ss-SERS shows a substantial enhancement of the SNR over 10 times at a modulation frequency of 250 MHz compared to 250 kHz. The experimental results further demonstrate the SNR comparison between the ss-SERS and signal averaging methods (Fig. 3d). The SNRs of R6G at 5 mM are compared between the ss-SERS signals for the 3,937,007 repetition times for code lengths of 64, 128, 256, and 512 and the signal averaging signals with an interval of 10 ms for the total measurement time. For this experiment, the maximum code length was set by 512 bits by considering the pixel number and the read-out rate of the spectrometer. Additionally, the maximum modulation frequency was set by 250 MHz by considering the effects of the cross-correlation of the fluorescence removal and the bit duration of the SERS signal intensity. Moreover, the PN codes of ss-SERS were selected from the primitive polynomials corresponding to the code lengths listed in Supplementary Fig. 5a.

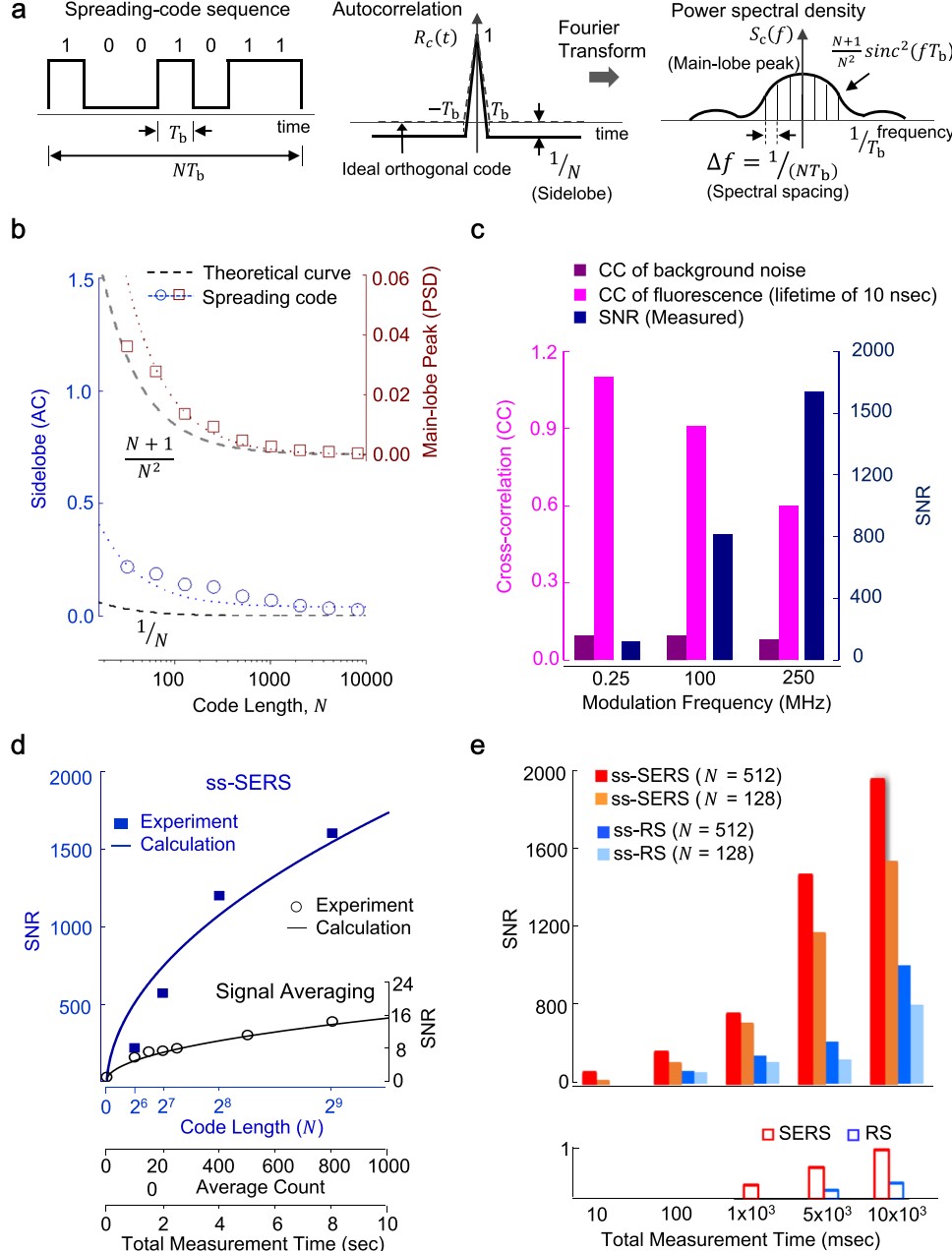

**Fig. 3 The signal-to-noise ratio of ss-SERS depends on the code length of a PN code and the modulation frequency. a** Conceptual description of the autocorrelation and power spectral density of the spreading codes as high orthogonal codes based on primitive polynomials. $T_b$, $N$, $R_c(\tau)$, and $S_c(\tau)$ represent a bit duration, code length, autocorrelation function, and power spectral density function, respectively. **b** The autocorrelation (AC) sidelobe and the main-lobe peak dependence on the PN code length. The autocorrelation sidelobe and the main-lobe peak for the high orthogonal code decrease to the theoretical level as the code length increases, resulting in significant noise suppression. **c** The cross-correlation (CC) and the measured SNRs of ss-SERS for R6G molecules at 1331 cm$^{-1}$ dependence on the modulation frequency. The cross-correlation of the fluorescence generated by excitation light decreases with increasing modulation frequency, however, the cross-correlation for the background noise independent of the excitation light remains near-zero. **d** Comparison of the measured SNRs between the ss-SERS and the signal averaging for R6G molecules at 1331 cm$^{-1}$. The total measurement time corresponds to the code length multiplied by a constant sequence repetition and bit duration as well as the average count multiplied by a constant sweep time. The ss-SERS measurement shows an exceptional improvement in the SNR by over two orders of magnitude compared to the signal averaging of the SERS signals. **e** Comparison of the temporal resolution between ss-SERS/ss-RS and conventional SERS/RS. The ss-SERS exhibits a substantial reduction in the measurement time over 100th compared to conventional SERS and RS.

The measured SNR of ss-SERS and the signal averaging signals are well matched with the theoretical coding gain curve of Eq. (6) (see "Methods" section for SNR calculation of the ss-SERS). Notably, the ss-SERS signals show an exceptional improvement in the SNR compared to the signal averaging of SERS signals. For instance, the SNR of ss-SERS signal is over two orders of magnitude greater

than that of the signal averaging for an 8 s measurement time. The suppression mechanism of fluorescence signals at a modulation frequency of 250 MHz can be explained in Supplementary Fig. 6 and the comparison between the ss-SERS and modulated-based SERS is summarized in Supplementary Table 2. The peak autocorrelation of the spreading code completely restores the

SERS signal from noisy signals because SERS signals with lifetimes of several picoseconds are encoded in the same pattern as the spreading code. The fluorescence with a lifetime of several nanoseconds to several milliseconds is encoded in a distorted pattern from the spreading code and then is substantially suppressed by the near-zero cross-correlation between the fluorescence noise and the spreading code. The signal averaging can theoretically improve the SNR up to the square root of the number of iterations under conditions where the signal and noise are uncorrelated and the noise is random and has a mean value of zero (see "Methods" section for SNR calculation of signal averaging and Eq. (13)). Consequently, the ss-SERS increases the SNR by more than two orders of magnitude compared to SERS in even a single measurement without signal averaging due to the efficient suppression of the fluorescence and optical fluctuation noise. The temporal resolutions between the ss-SERS/ss-RS and the SERS/RS were experimentally compared in Fig. 3e. The ss-SERS method allows the measurement of the peak intensities for 5 mM R6G molecules even at a short time of 10 ms, whereas the measurable time of SERS is still limited to 1 s. A high temporal resolution is also observed for ss-RS; for instance, ss-RS measures the peak intensities for 5 mM R6G molecules even at a minimum of 100 ms. However, the RS method hardly distinguishes the Raman spectra for <5 s. Consequently, compared to conventional SERS and RS methods, both ss-SERS and ss-RS methods exhibit a substantial reduction in the measurement time by a factor of 100.

**Label-free detection of primary neurotransmitters at attomolar labels**. The SNR enhancement of ss-SERS allows the label-free detection of primary neurotransmitters, such as dopamine, serotonin, acetylcholine, GABA, and glutamate (Fig. 4). The characteristic Raman peaks for neurotransmitters at 1 mM

concentrations were clearly distinguished by using ss-SERS (Supplementary Figs. 7–11). Note that the Raman vibrations in these spectra are barely measurable with SERS due to the low Raman activity of neurotransmitters. A characteristic Raman peak combination for the selective detection of primary neurotransmitters was further extracted by using a PCA (Supplementary Fig. 12). The intensity of characteristic Raman peaks exhibits higher linearity with the sample concentration as well as a wider dynamic range than those of other peaks. The experimental results indicate that dopamine at 1402 cm$^{-1}$ (C–H wagging and N–H twisting), serotonin at 927 cm$^{-1}$ (out of phase breathing), acetylcholine at 1150 cm$^{-1}$ (CH$_3$ rocking and CH$_2$ wagging), GABA at 1260 cm$^{-1}$ (CH$_2$ wagging), and glutamate at 1114 cm$^{-1}$ (CH$_2$ rocking), are clearly three orders of magnitude higher than those of SERS. Such an exceptional SNR enhancement of ss-SERS allows the clear detection of major ss-SERS peak intensities for major neurotransmitters at an ultralow concentration level (Supplementary Figs. 13 and 14). For instance, the ss-SERS spectra of acetylcholine for different concentrations ranging from 1 mM ($10^{-3}$ M) to 1 aM ($10^{-18}$ M) are presented in Fig. 4a. The acetylcholine molecules are associated with ss-SERS peaks at 776, 927, 1114, 1150, and 1331 cm$^{-1}$ and are readily detected at a single attomole level, particularly the 1150 cm$^{-1}$ peak. Finally, ss-SERS clearly demonstrates the LOD for primary neurotransmitters at attomolar levels (see "Methods" section for LOD measurement for primary neurotransmitters; Supplementary Table 3): acetylcholine of the learning neurotransmitter in saline solution shows an LOD of 1 aM (Fig. 4b). In addition, GABA of the calming neurotransmitter, serotonin of the mood neurotransmitter, glutamate of the memory neurotransmitter, and dopamine of the pleasure neurotransmitter display LODs of 99, 3, 39, and 1.9 aM, respectively. (Fig. 4b; Supplementary Figs. 15, 16). The ss-SERS method allows the minimum LOD for

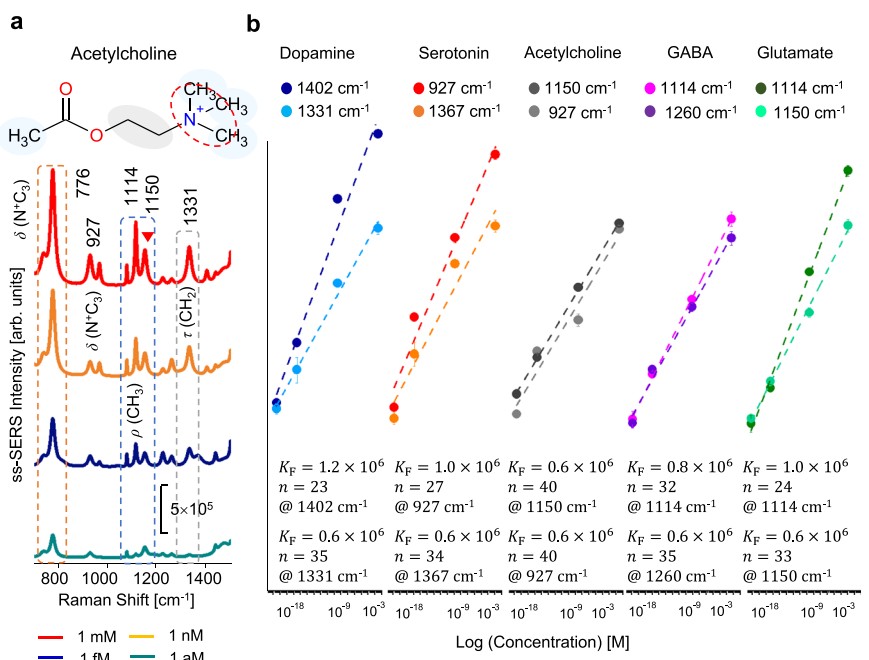

**Fig. 4 LODs and ss-SERS spectra of primary neurotransmitters for different concentrations ranging from 1 mM ($10^{-3}$ M) to 1 aM ($10^{-18}$ M). a** The characteristic Raman peak intensities of ss-SERS for acetylcholine of the learning neurotransmitter associated with Alzheimer's dementia depending on neurotransmitter concentrations. The ss-SERS spectra exhibit a major SERS peak at 1150 cm$^{-1}$ assigned to CH$_3$ rocking and CH$_2$ wagging. $\delta$ bending, $\tau$ twisting, $\omega$ wagging, $\nu$ stretching, $\rho$ rocking. (The output power of the laser: 25 mW, the power at the sample: 1 mW, accumulation time: 10 s.) **b** Limit of detection for primary neurotransmitters down to attomolar concentrations. The nonlinear fit curves based on the experimental data for the neurotransmitters agree well with the Freundlich isotherm-like behavior ($\log q_e = \log K_F + \frac{1}{n} \log C_e$), where $q_e$, $K_F$, $n$, and $C_e$ represent the ss-SERS peak intensity, Freundlich isotherm constant, Freundlich isotherm exponent, and concentration, respectively.

**Table 1 LOD and SNR enhancement of ss-SERS.**

| Neurotransmitters | Dopamine @1402 cm$^{-1}$ | Serotonin @927 cm$^{-1}$ | Acetylcholine @1150 cm$^{-1}$ | GABA @1260 cm$^{-1}$ | Glutamate @1114 cm$^{-1}$ |
|---|---|---|---|---|---|
| LOD with ss-SERS | 1.9 aM | 3.0 aM | 1.0 aM | 99 aM | 39 aM |
| Measured SNR EF (SNR$_{ss\text{-}SERS}$/SNR$_{SERS}$) @ 1 mM | $1.9 \times 10^3$ | $1.3 \times 10^3$ | $1.5 \times 10^3$ | $0.5 \times 10^3$ | $0.8 \times 10^3$ |
| Reported LOD with SERS[27,31] | 600 nM | 100 nM | 4.0 µM | 50 µM | 600 nM |

neurotransmitters reported to date, whereas conventional label-free SERS detection typically obtains an LOD from 1 µM to several nM[27,31]. The LODs and SNR enhancement ratio obtained with ss-SERS are summarized and compared with those of conventional methods in Table 1. The inverse relation between the LOD and SNR enhancement is clear evidence that the remarkable SNR enhancement of ss-SERS substantially contributes to lowering the LOD.

## Discussion

To conclude, this work has successfully demonstrated the ultrasensitive ss-SERS detection of neurotransmitter molecules with extremely low Raman activities using spreading coded light excitation. Fluorescence signals distorted or background noises unrelated to the spreading code of the encoder are completely filtered out by using peak autocorrelation and near-zero cross-correlation. The experimental results have clearly demonstrated that ss-SERS substantially increases the SNR by more than three orders of magnitude and reduces the high temporal resolution by more than 100 compared to conventional SERS. As a result, ss-SERS clearly shows label-free and attomolar detection of primary neurotransmitters with extremely low Raman activities, such as dopamine, serotonin, acetylcholine, GABA, and glutamate. These experimental results provide a diagnostic method for investigating the early diagnostics of neurological disorders or highly sensitive biomedical SERS applications. Moreover, spreading coded light excitation can provide a route for developing highly sensitive, high speed, low cost, and handheld biomedical spectroscopic techniques in UV, visible, infrared, and even terahertz ranges.

## Methods

**Sample preparation.** Rhodamine 6G (R6G) powder, 3-hydroxytyramine hydrochloride (dopamine, 99%), 5-hydroxytrytamine hydrochloride (serotonin), acetylcholine chloride (99%), gamma-Amino-n-butyric acid crystalline (GABA), and glutamic acid monosodium salt monohydrate were purchased from Sigma-Aldrich. The liquid sample of R6G ($C_{28}H_{31}ClN_2O_3$) at 5 mM concentration as a reference molecule for SERS measurements was prepared by mixing powder and distilled water. The R6G is a highly fluorescent Rhodamine family dye with high Raman activity, consisting of a xanthene ring, two $NHC_2H_5$ groups, two methyl groups, and a phenyl ring with $COOC_2H_5$ groups. The liquid samples of primary neurotransmitters, i.e., dopamine, serotonin, acetylcholine, GABA, and glutamate, were diluted in saline solution to four concentrations of 1 aM, 1 fM, 1 nM, and 1 mM. Dopamine ($C_8H_{11}NO_2$) as a pleasure neurotransmitter is associated with feelings of pleasure, satisfaction, and addiction, and the loss of dopamine-containing neurons in the midbrain progresses in PD. Serotonin ($C_{10}H_{12}N_2O$) as a mood neurotransmitter is related with feelings of well-being and happiness and regulates the sleep cycle and intestinal movements. The intense research in biological psychiatry has demonstrated serotonin's influence on depression and suicide, as well as on the pathogenesis of AD. Acetylcholine ($C_7NH_{16}O_{2+}$) is a principal neurotransmitter involved in thought, learning and memory, which has the particular ability to bind to both nicotinic and muscarinic receptors. Gamma-aminobutyric acid (GABA, $C_4H_9NO_2$) is a neurotransmitter that increases the permeability of the postsynaptic membranes to $K^+$, acting as a powerful inhibitor of the synaptic transmission. Increased levels improve mental focus and relaxation. Glutamate ($C_5H_9NO_4$) as the major excitatory neurotransmitter in the human brain is involved in many neuronal functions including synaptic transmission, neuronal migration, excitability and long-term potentiation.

**SERS substrate preparation.** The Au nanoislands on quartz glasses were prepared for the experiment as a SERS substrate. The gold was simply coated (10 nm) on a quartz glass through thermal evaporation of Au thin film. The coated Au thin film in Volmer–Weber mode directly forms the nanoislands on the top surface of the quartz glass due to the strong coupling of Au atoms with each other. Dedicate control of film thickness and deposition rate enables formation of Au nanoislands in Volmer–Weber node. Extinction spectra were calculated as $1 - R_{np}/R_g$ after the intensity values of reflected light intensity $R$ from the nanoplasmonic glass ($R_{np}$) and quartz glass ($R_g$) were measured using a charge-coupled device (CCD)-based UV–vis near-infrared (NIR) micro-spectrometer (SpectraPro 2300i, Princeton Instruments) coupled with an inverted confocal laser scanning microscope (CLSM, Axiovert 200M).

**SNR calculation of the ss-SERS.** We derive the equation for SNR gain of the ss-SERS. A PN sequence modulating a laser is denoted as $f(t)$ with a unit amplitude, a bit period $T$ and a bit number $N$. The scattered sequence, corresponding to the encoded Raman signals, from the target molecule will be $f(t - iT)$, where $s$ represents the scattered fraction and $f(t - iT)$ is a delayed PN sequence by $i$ bits. After combining the delayed PN sequence, $sf(t - xT)$ with the encoded Raman signals, $sf(t - iT)$ in the integration time $NT$, the correlator output[43,44].

$$C(x) = \frac{1}{NT}\int_0^{NT} f(t - xT)sf(t - iT)\mathrm{d}t. \qquad (1)$$

From the correlation property of a PN signal:

$$\frac{1}{NT}\int_0^{NT} f(t - aT)f(t - bT)\mathrm{d}t \approx \begin{cases} 1 & a = b \\ -1/N & a \neq b \end{cases} \qquad (2)$$

We have

$$C(x) = \frac{s}{NT}\int_0^{NT} f^2(t - xT)\mathrm{d}t\bigg|_{i=x} + \frac{s}{NT}\int_0^{NT} f(t - xT)f(t - iT)\mathrm{d}t\bigg|_{i\neq x}, \qquad (3)$$

$$\approx s - \frac{1}{N}\bar{s}. \qquad (4)$$

The noises are significantly suppressed because the second term is close to zero when the value of $N$ becomes larger. The $\bar{s}$ is the mean value of interference signals such as non-specific binding and broad fluorescence noise that are not equal to the period of the encoder's PN sequence. On the other hand, system noise arising from the system itself can be divided into two parts, optical noise and non-optical noise. Optical noise is the measurement error originating from incident light such as laser power variation and laser frequency drift. Non-optical noise is the measurement error independent of the incident light such as shot noise, thermal noise, and dark current. Only half of the codes are in the high levels in a PN sequence and so have optical noise. Thus, if $n_o$ represents the root mean square (RMS) of optical noise in one bit, adding the optical noise of whole sequence gives $n_o\sqrt{N/2}$ and the mean optical noise after integration becomes.

$$\frac{1}{N}n_o\sqrt{N/2} = n_o\frac{1}{\sqrt{2N}}. \qquad (5)$$

If $n_n$ represents the RMS of non-optical noise in one bit, integration over the whole PN sequence gives $n_n/\sqrt{N}$. The SNR of the ss-SERS (SNR$_c$) can be deduced as

$$\mathrm{SNR}_c = \frac{s}{\frac{1}{N}\bar{s} + \frac{1}{\sqrt{2N}}n_o + \frac{1}{\sqrt{N}}n_n} = \frac{\mathrm{SNR}_b}{\frac{1}{N}\mathrm{SNR}_b + \frac{1}{\sqrt{2N}}\frac{n_o}{n_o + n_n} + \frac{1}{\sqrt{N}}\frac{n_n}{n_o + n_n}}, \qquad (6)$$

where

$$\mathrm{SNR}_b = \frac{s}{n_o + n_n} \approx \frac{\bar{s}}{n_o + n_n} \qquad (7)$$

is the mean SNR in one bit without the correlation effect. It is clear from Eq. (6) that the SNR increases as the sequence is longer.

**SNR calculation of signal averaging.** Conventional method to increase the SNR is averaging data. It is the random nature of noise that makes signal averaging useful, that is, the assumptions are that the signal and noise are uncorrelated and that the noise is random and has a mean value of zero. The output of single SERS

measurement $f(t)$ has a SERS signal portion $S(t)$ and a noise portion $N(t)$. Then

$$f(t) = S(t) + N(t). \qquad (8)$$

If $f(t)$ is sampled every $T$ seconds, the value of any sample point is the sum of the noise component and signal component.

$$f(iT) = S(iT) + N(iT). \qquad (9)$$

The value after $m$ repetitions is

$$\sum_{i=1}^{m} f(iT) = \sum_{i=1}^{m} S(iT) + \sum_{i=1}^{m} N(iT). \qquad (10)$$

The signal component for sample point $i$ is the same at each repetition if the signal is stable. Then

$$\sum_{i=1}^{m} S(iT) = mS(iT). \qquad (11)$$

After many repetitions, $N(iT)$ has a rms value of $\sigma_n$.

$$\sum_{i=1}^{m} N(iT) = \sqrt{m\sigma_n^2} = \sqrt{m}\sigma_n. \qquad (12)$$

The SNR after $m$ repetitions ($\text{SNR}_m$) is deduced as

$$\text{SNR}_m = \frac{\sum_{i=1}^{m} S(iT)}{\sum_{i=1}^{m} N(iT)} = \frac{mS(iT)}{\sqrt{m\sigma_n^2}} = \sqrt{m}\,\text{SNR}. \qquad (13)$$

Thus, signal averaging improves the SNR by a factor of $\sqrt{m}$.

**LOD measurement for primary neurotransmitters**. The measured LODs are obtained as the intercept between the nonlinear fit curve based on the experimental data, and the line corresponding to the value of a blank sample (0 M) plus three times its standard deviation noise. The nonlinear fit curves for the neurotransmitters agree well with the Freundlich isotherm-like behavior. The Freundlich isotherm assumes that the adsorbates are adsorbed onto the heterogeneous surface of an adsorbent while the Langmuir isotherm assumes monolayer adsorption on a uniform surface with a finite number of adsorption sites. $K_F$ is an indicator of adsorption capacity, so higher the maximum capacity, higher the $K_F$. The isotherms present a linear relationship between SERS intensity and concentrations of the neurotransmitters.

**Reporting summary**. Further information on research design is available in the Nature Research Reporting Summary linked to this article.

## Data availability
The authors declare that the main data supporting the findings of this study are available within the article and its Supplementary Information file. Extra data are available from the corresponding author upon request.

## Code availability
The authors declare that the main algorithms of the custom codes used in this study are fully explained in the article and its Supplementary Information file. All codes used for analysis of this study are also available from the corresponding authors upon reasonable request.

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

## Acknowledgements

We thank J.S. Cho for the useful discussion of the spread-spectrum technique used in telecommunication. This work was financially supported by the National Research Foundation of Korea (MSIT Nos. 2016R1A2B301306115 and 2016M3A9B691919322), the KAIST Mobile Clinic Module Project (MSIT No. MCM-2020-N11200215), the Korea Medical Device Development Fund (MSIT/MTIE/MOHW/MFDS No. 202011D11), and Institute for Information & Communications Technology Promotion (IITP) (MSIT No. 18HS1960) funded by the Korean government.

## Author contributions

W.L. and K.-H.J. directed the study. M.P. provided the SERS papers, J.H.K. and B.-H.K., and H.Y. prepared the R6G and neurotransmitter samples. W.L. performed all experiments, data processing, and analysis. B.K.K., T.C., and Y.J. assisted with the useful discussions of the experimental results. W.L. and K.-H.J. wrote the manuscript with contributions from the other authors. K.-H.J. supervised the work.

## Competing interests

The authors declare no competing interests.
