## [Peer Review File · Nature Communications]

Reviewer #1 (Remarks to the Author):

The manuscript by Lee et al presents a new method for removing background signal and fluorescence from the pure SERS signal. The authors state that their methodology allows for detection of neurotransmitters to the attomolar concentration range. While this methodology and results may be of interest to researchers in the field of SERS, there may be limited interest for a wider audience. Before this manuscript can be recommended for publication, there are several issues which would need to be addressed by the authors. These are outlined below:

(1) An elegant and complicated method for post-processing of the SERS data is presented, however, the multi-step processing method allows for introduction of spurious, somewhat odd peaks in the spectra. These lead to some concern. Specifically, in the ss-SERS spectrum of R6G in Figure 2c, there are a large number of extra peaks where the origin of the peaks is not clear. Additionally, in Figure SI12, when comparing the ss-SERS spectra at 1 fM and 1 aM, there are some peaks where the intensity changes between concentrations do not make sense. For example, the high intensity peak at ~ 800 cm^{-1} (highlighted with the orange bar) has two peaks on either side of it. Looking at the higher cm^{-1} peak, the relative change in intensity from 1 fM to 1 aM is not consistent. At ~ 1050 cm^{-1} , there is a high intensity peak in the 1 aM spectrum that is not present in the 1 fM spectrum. Finally, there are several small peaks where the intensity is similar between all 4 concentrations. Granted these are not peaks of interest, however, the behaviour between them at the different concentrations is not consistent and causes some concern that these are arising from the processing of the data. These patterns of peaks are also seen in the ss-SERS spectra in Figures SI5, SI6, SI7, SI8, SI9, and SI10. The authors should see if they can explain these peaks and how they are confident that the processing is not adding noise/peaks to the spectra.

(2) Page 4, lines 65-66: the authors mention that SERS can be used for neurotransmitter sensing, however the cited reference from Le Ru and Etchegoin (26) is about single molecule SERS, not specifically neurotransmitters. Additionally, there are at least two recent publications on SERS of neurotransmitters that have not been cited by the authors, including Ashley et al, *J. Phys. Chem. C*, 2018 122, 2307-2314 and Moody et al., *ACS Chemical Neuroscience*, 2018, 9, 6, 1380-1387. Both publications demonstrate SERS detection down to the 100 nM concentration range. These should be cited.

(3) Page 4, lines 79-81: the authors state that SERS substrates require high cost top-down methods, but many groups have demonstrated reliable SERS measurements using colloidal nanoparticles (for example the groups of Duncan Graham and Karen Faulds at the University of Strathclyde). This statement should be amended.

(4) For the spectra in Figures 2, 4, SI5, SI6, SI7, SI8, SI9, SI10, SI12, the power at the sample and accumulation time should be included. In the manuscript at power of 25 mW is mentioned, which seems to be high for a fiber coupled system.

(5) Figure 4C: technically this should be a table, not part of a figure. Also, for the last line of the table, the "reported LOD with SERS," are these values from the literature? If so, references need to be included. The values are also not reflective of the lowest LODs determined with SERS, the papers mentioned above in comment (2) outline lower LODs.

Reviewer #2 (Remarks to the Author):

Generally speaking, I am very excited about this article and about its approach; I believe that using these sorts of signal-processing techniques are of critical need for this type of spectroscopy. The results are impressive. I predict that this approach will be utilized extensively across the SERS community, and may trigger additional activity and applications of this type of spectroscopy. I do believe that the authors need to take care of several issues before this work is ready for acceptance:

1. There are a number of minor spelling and grammatical mistakes (e.g. on line 15 it should be "...other neurons by releasing..."). I've called out a few here, but there are many places in the text where a grammatical check would be helpful.
 - a. Line 100: "...(OTDR), except for biomedical sensing applications." I don't follow what is meant by "except" here.
 - b. In SI Table 5, K_f and n are labelled as Freundlich (note the m instead of n)
2. It would be useful if the authors could describe in more detail what is meant by (Line 163-164) "The convolution result is transformed into the conventional Raman spectrum of Lorentzian line shape using intensities, center positions, and standard deviations of the decoded SERS peaks...". In the SI they describe some of the mathematics that goes in to this process, but it is not clear to me if it is a fitting procedure to identify these quantities (intensity,...) or if they are using previously-known literature values (which does not appear to be the case).
3. Continuing with the above question, why is this final transform even necessary? Following Step 6 of Sup. Fig. 2, the result is the noise-free SERS signal as a function of position on the spectrometer, which can be directly converted into a measured spectrum. Indeed, this seems to be what is shown in Fig. 2b (top), so why is this final processing step necessary? Is it just to extract the various parameters for each peak? It seems there is more going on as the peak (in the ss-SERS data) at 1368 cm^{-1} in the top panel is highly asymmetric (by eye) while the plotted spectrum shows it to be perfectly symmetric. More detail about this process would be useful, and potentially a comparison in the SI of the raw data after processing step 6 with the final output spectrum after the Lorentzian transformation.
4. Finally, to finish this line of questioning, why is it that all the experimental peaks are perfectly Lorentzian in shape? I assume this is related to the aforementioned transform, but it strikes me as very odd that the results would be so perfectly smooth. Even with the impressive approach described in the paper I would still expect some amount of noise to be evident in the Raman spectra, or at least that they not always be so shaped so precisely.
5. The discussion on the calculation of the LOD is missing important details – particularly because it is not obvious from Fig. 4b how the plotted curves lead to the reported LOD. For example, serotonin and GABA are claimed to have identical LOD, but the plotted curves have different y -intercepts and by eye similar amounts of noise (both have 2 large error bars, 1 medium, and 1 small), but are listed as having the same LOD. The SI describes the process for finding the LOD but leaves out the values used for the 0 M concentration lines and its associated noise.

Title: “Spread Spectrum SERS allows label-free detection of attomolar neurotransmitters”

Wonkyoung Lee^{1, 2, 3}, Byoung-Hoon Kang^{1, 2}, Hyunwoo Yang^{1, 2}, Moonseong Park¹, Ji Hyun Kwak¹, Taerin Chung¹, Yong Jeong^{1, 2}, Bong Kyu Kim³, and Ki-Hun Jeong^{1, 2*}

¹Department of Bio and Brain Engineering, Korea Advanced Institute of Science and Technology (KAIST), 291 Daehak-ro, Yuseong-gu, Daejeon 305-701, Republic of Korea

²KAIST Institute for Health Science and Technology (KIHST), KAIST, 291 Daehak-ro, Yuseong-gu, Daejeon 305-701, Republic of Korea

³Bio-Medical IT Convergence Research Division, Electronics and Telecommunications Research Institute (ETRI), 218 Gajeong-ro, Yuseong-gu, Daejeon, 350-700, Republic of Korea

*Corresponding author: kjeong@kaist.ac.kr

Author’s response for the Reviewer’s comments:

First of all, we deeply appreciate the reviewers for their interest in our manuscript and their constructive comments. This work reports the first demonstration of spreading spectrum light excitation technique for ultrasensitive SERS applications; Under laser excitation of spreading code, SERS signals are encoded with the spreading code and fully decoded into quasi-noise-free SERS signals after the complete removal of all the background noise, including fluorescence. As the reviewer pointed out, we strongly believe that this technique can be widely and effectively utilized for highly sensitive and low-cost spectroscopic applications such as SERS, RS, UV-VIS, FTIR, SPR, LSPR or even THz TDS.

The main concerns of two reviewers are regarding the shape of the spectra after the multi-step processing. i.e., What is the cause of a large number of extra peaks and inconsistent peaks after the multi-step processing? and why is this final processing step (Lorentzian line shape transformation) necessary?

Based on the comments from the first reviewer, we have performed additional experiments and found out the cause of extra peaks and inconsistent peaks. It turns out that the cause of the extra peaks and inconsistent peaks is originated from quartz substrate. In other words, ss-RS and ss-SERS peaks of the substrate correspond to vibrational bands of the Si-O stretching modes and Au nanoislands in the quartz glass and plasmonic glass. By using the calibration step that subtracts ss-SERS signals of the substrate itself from the ss-SERS spectrum of the sample on the substrate, we confirm that in the calibrated spectra, there are no spurious or extra peaks and the intensities of the ss-RS and ss-SERS peaks consistently change according to the concentration. With additional new experiments, we have revised all the data (Fig. 2, 4, SI Fig. 7, 8, 9, 10, 11, 13, and 16) using the calibrated spectra.

Based on the comments from the second reviewer, we have faithfully explained the necessity of final processing step (Lorentzian line shape transformation) and more detail description of the Lorentzian line shape transformation (in the figure below). We emphasized that the accurate line shape fitting of the Raman spectrum is even more important, particularly for low-resolution spectrometers, in order to extract the line widths of Raman peaks indicating the crystallinity of material and the influence of various environment. And then we have carefully revised the LOD values of the neurotransmitters with the calibrated ss-SERS spectra and added important details in LOD calculation such as the values used for the zero molar concentration and its associated noise in SI Fig. 15. In addition, for selective detection of primary neurotransmitters, we have newly added extraction of a characteristic Raman peak combination for each neurotransmitter that does not overlap between neurotransmitters by using the principal component analysis (PCA) tool known as dimensionality-reduction method for feature extraction. We confirmed that the characteristic Raman peaks have higher linearity in intensity versus concentration and wider linear dynamic range than those of other peaks.

Including the main concern, we have faithfully addressed the reviewer's point-by-point comment in the attached author response as well as added more detail description in the revised manuscript as follows.

The grammatical issues were resolved by Springer Nature Author Service (SNAS). The highly qualified native English speaking editors of Springer Nature Author Service (SNAS) edited the manuscript in the proper English language, punctuation, spelling and overall style to improve flow and readability of the text.

SPRINGER NATURE

Author Services

Editing Certificate

This document certifies that the manuscript

Spread Spectrum SERS allows label-free detection of attomolar neurotransmitters

prepared by the authors

Wonkyoung Lee, Byoung-Hoon Kang, Hyunwoo Yang, Moonseong Park, Ji Hyun Kwak, Taerin Chung, Yong Jeong, Bong Kyu Kim, and Ki-Hun Jeong

was edited for proper English language, grammar, punctuation, spelling, and overall style by one or more of the highly qualified native English speaking editors at SNAS.

This certificate was issued on **August 3, 2020** and may be verified on the SNAS website using the verification code **637F-B6BC-12C0-A7F9-E13P**.

Neither the research content nor the authors' intentions were altered in any way during the editing process. Documents receiving this certification should be English-ready for publication; however, the author has the ability to accept or reject our suggestions and changes. To verify the final SNAS edited version, please visit our verification page at secure.authorservices.springernature.com/certificate/verify. If you have any questions or concerns about this edited document, please contact SNAS at support@as.springernature.com.

SNAS provides a range of editing, translation, and manuscript services for researchers and publishers around the world. For more information about our company, services, and partner discounts, please visit authorservices.springernature.com.

For the 1st Reviewer's comments

Detail address:

1.1) Reviewer's comment:

The manuscript by Lee et al presents a new method for removing background signal and fluorescence from the pure SERS signal. The authors state that their methodology allows for detection of neurotransmitters to the attomolar concentration range. While this methodology and results may be of interest to researchers in the field of SERS, there may be limited interest for a wider audience. Before this manuscript can be recommended for publication, there are several issues which would need to be addressed by the authors. These are outlined below:

An elegant and complicated method for post-processing of the SERS data is presented, however, the multi-step processing method allows for introduction of spurious, somewhat odd peaks in the spectra. These lead to some concern. Specifically, in the ss-SERS spectrum of R6G in Figure 2c, there are a large number of extra peaks where the origin of the peaks is not clear.

Author's response:

We confirmed that the spurious peaks shown in the spectra are not peaks caused by the multi-step processing, but peaks corresponding to ss-RS and ss-SERS signals of substrate materials, i.e., quartz glass and plasmonic glass of Au nanoislands on quartz glass, by measuring ss-RS and ss-SERS of substrates and comparing to spectra of R6G on the substrates.

We removed ss-RS and ss-SERS peaks of quartz glass and plasmonic glass by subtracting the ss-RS and ss-SERS signals of substrates from the ss-RS and ss-SERS spectra of R6G on the substrates. In the calibrated spectra, the spurious and extra peaks pointed out by the reviewer are not shown, and the intensities of the ss-RS and ss-SERS peaks consistently change according to the R6G concentration. In addition, we confirmed that ss-RS and ss-SERS peaks of the substrate correspond to vibrational bands of the Si-O stretching modes and Au nanoislands in the quartz glass and plasmonic glass in the literatures^{1,2}.

In order to clarify ss-SERS spectra of R6G, the authors revised manuscript by adding steps to obtain calibrated spectra and revised Figure 2b and Figure 2c with the calibrated spectra, where all the revised texts were high-lighted in red. The authors also corrected some words and phrases.

Page. Column	Original	Author's Correction
P8.9.		The decoded SERS peaks of the target molecules are clearly obtained by subtracting the ss-SERS signals of a substrate material from the restored SERS signals (Supplementary Fig. 2).
P20. Figure 2b.		

P20.
Figure 2c.

The revised supporting information is attached below.

Supplementary Figure 2 is added in supporting information

Supplementary Figure 2. The calibrated ss-RS and ss-SERS spectra. (a) The calibrated ss-RS spectra are obtained by subtracting the ss-RS signals of quartz glass from the ss-RS signals of R6G on quartz glass. (b) The calibrated ss-SERS spectra are obtained by subtracting the ss-SERS signals of plasmonic glass from the ss-SERS signals of R6G on plasmonic glass.

1.2) Reviewer's comment:

Additionally, in Figure SI12, when comparing the ss-SERS spectra at 1 fM and 1 aM, there are some peaks where the intensity changes between concentrations do not make sense. For example, the high intensity peak at $\sim 800\text{ cm}^{-1}$ (highlighted with the orange bar) has two peaks on either side of it. Looking at the higher cm^{-1} peak, the relative change in intensity from 1 fM to 1 aM is not consistent. At $\sim 1050\text{ cm}^{-1}$, there is a high intensity peak in the 1 aM spectrum that is not present in the 1 fM spectrum. Finally, there are several small peaks where the intensity is similar between all 4 concentrations.

Author's response:

As shown in Supplementary Figure 2, extra peaks in ss-SERS spectra are originated from the Raman signals of the quartz glass and plasmonic glass themselves. In addition, the surface conditions (i.e. defective, cluster of Au island, cleanness) of the quartz glass and plasmonic glass are different depending on the position of wafer, so the relative change in intensity for some peaks is not consistent. For instance, the peak intensity of 1 aM at 737 cm^{-1} is higher than that of 1 fM, and there is a high intensity peak at 1039 cm^{-1} in the 1 aM spectrum that is not present in the 1 fM spectrum. In the same process as the calibration for R6G in Figure 2, the authors removed ss-RS and ss-SERS peaks of quartz glass and plasmonic glass by subtracting the ss-RS and ss-SERS signals of the substrates from the ss-RS and ss-SERS spectra of dopamine in Supplementary Figure 16.

In the calibrated spectra, the relative change in intensity at 737 cm^{-1} from 1 fM to 1 aM is consistent. The peak at 1039 cm^{-1} in the 1 aM spectrum that is not present in the 1 fM spectrum disappears. Also, several small peaks where the intensity is similar between all 4 concentrations disappear. The relative intensities for ss-RS and ss-SERS peaks change consistently with concentrations by removing ss-RS and ss-SERS peaks of quartz glass and plasmonic glass.

The revised supporting information is attached below.

Supplementary Figure S16 in supporting information

(original)

(Author's correction)

1.3) Reviewer's comment:

Granted these are not peaks of interest, however, the behavior between them at the different concentrations is not consistent and causes some concern that these are arising from the processing of the data. These patterns of peaks are also seen in the ss-SERS spectra in Figures SI5, SI6, SI7, SI8, SI9, and SI10. The authors should see if they can explain these peaks and how they are confident that the processing is not adding noise/peaks to the spectra.

Author's response:

In summary, inconsistent peaks and small extra peaks were found to be caused by ss-RS and ss-SERS signals of the quartz glass and plasmonic glass. The authors obtained clear spectra that peak intensities change consistently according to concentration and there are no extra peaks unrelated to target molecules by subtracting ss-RS/ss-SERS signals of substrates themselves. Therefore, the authors are confident that the processing don't add noise/peaks to the spectra.

From Supplementary Figure 7 to Supplementary Figure 10 are revised by the same calibration process as Figure 2.

The revised supporting information are attached below.

Figure	Original	Author's Correction
SI-Figure 7		
SI-Figure 8		

SI-Figure 9

SI-Figure 10

SI-Figure 11

SI-Figure 13

1.4) Reviewer's comment:

Page 4, lines 65-66: the authors mention that SERS can be used for neurotransmitter sensing, however the cited reference from Le Ru and Etchegoin (26) is about single molecule SERS, not specifically neurotransmitters. Additionally, there are at least two recent publications on SERS of neurotransmitters that have not been cited by the authors, including Ashley et al, J. Phys. Chem. C, 2018 122, 2307-2314 and Moody et al., ACS Chemical Neuroscience, 2018, 9, 6, 1380-1387. Both publications demonstrate SERS detection down to the 100 nM concentration range. These should be cited.

Author's response:

According to the reviewer's comment, the authors revised the manuscript by adding the references.

Page	Original	Author's Correction
P16. References		(26) Ashley, M. J. et al. Shape and Size Control of Substrate-Grown Gold Nanoparticles for Surface-Enhanced Raman Spectroscopy Detection of Chemical Analytes. The Journal of Physical Chemistry C 122, 2307-2314 (2018). (27) Amber S. Moody, B. S. Multi-metal, Multi-wavelength Surface-Enhanced Raman Spectroscopy Detection of Neurotransmitters. ACS Chem Neurosci 9, 1380-1387 (2018).

1.5) Reviewer's comment:

Page 4, lines 79-81: the authors state that SERS substrates require high cost top-down methods, but many groups have demonstrated reliable SERS measurements using colloidal nanoparticles (for example the groups of Duncan Graham and Karen Faulds at the University of Strathclyde). This statement should be amended.

Author's response:

As considering the reviewer's comment, the authors corrected some words and phrases.

Page. Column	Original	Author's Correction
P4. 18	However, plasmonic SERS-active substrates require high-cost top down nanofabrication method to maintain reproducibility and stability of SERS signals for practical sensing applications.	However, plasmonic SERS-active substrates have mostly required high-cost top down nanofabrication methods or highly ordered nanoparticle synthesis to obtain stable and reproducible SERS signals for practical biochemical applications.

1.6) Reviewer's comment:

For the spectra in Figures 2, 4, SI5, SI6, SI7, SI8, SI9, SI10, SI12, the power at the sample and accumulation time should be included. In the manuscript at power of 25 mW is mentioned, which seems to be high for a fiber coupled system.

Author's response:

According to the reviewer's comment, the authors revised the manuscript by including the power at the sample and accumulation time in the captions of Figures 2, 4, Supplementary Figures 5, 6, 7, 8, 9, 10, and 12.

The power of 25 mW mentioned in the manuscript is the output power of laser considering the insertion loss of optical devices such as an electro-optic modulator and a polarization controller and the connection loss between optical fibers. Indeed, the power at the sample is about 1 mW, which I believe is an appropriate value for a fiber coupled system.

Page. Column	Original	Author's Correction
Figure 2b and Figure 4a captions, Supplementary Figure 5 ~ Supplementary Figure 10 captions, Supplementary Figure 12 caption		(The output power of the laser: 25 mW, the power at the sample: 1 mW, accumulation time: 10 sec)

1.7) Reviewer's comment:

Figure 4C: technically this should be a table, not part of a figure. Also, for the last line of the table, the "reported LOD with SERS," are these values from the literature? If so, references need to be included. The values are also not reflective of the lowest LODs determined with SERS, the papers mentioned above in comment (2) outline lower LODs.

Author's response:

According to the reviewer's comment, the authors separated Figure 4c into a figure and a table and added the papers mentioned in comment (2) as references to the table.

The revised figures are listed in the below.

Page. Column	Original	Author's Correction																																																
P.24 Figure 4	Acetylcholine Calibration Curves:  Dopamine: $y = 0.032x + 6.1$, $K_D = 1.4 \times 10^6$, $n = 31$ Serotonin: $y = 0.023x + 6.1$, $K_D = 1.2 \times 10^6$, $n = 43$ Acetylcholine: $y = 0.043x + 6.3$, $K_D = 2.1 \times 10^6$, $n = 23$ GABA: $y = 0.073x + 6.3$, $K_D = 2.1 \times 10^6$, $n = 13$ Glutamate: $y = 0.068x + 6.4$, $K_D = 2.4 \times 10^6$, $n = 14$ Table 1 (Original):    Neurotransmitters Dopamine Serotonin Acetylcholine GABA Glutamate     LOD with ss-SERS 25 aM 1.0 aM 2.1 aM 1.0 aM 10 aM   Measured SNR EF (SNR_{ss-SERS}/SNR_{SERS}) 1.6 × 10³ 2.0 × 10³ 2.6 × 10³ 3.0 × 10³ 2.6 × 10³   Reported LOD with SERS 10 nM 10 μM 1.0 μM 8.0 μM 100 nM   	Neurotransmitters	Dopamine	Serotonin	Acetylcholine	GABA	Glutamate	LOD with ss-SERS	25 aM	1.0 aM	2.1 aM	1.0 aM	10 aM	Measured SNR EF (SNR _{ss-SERS} /SNR _{SERS})	1.6 × 10 ³	2.0 × 10 ³	2.6 × 10 ³	3.0 × 10 ³	2.6 × 10 ³	Reported LOD with SERS	10 nM	10 μM	1.0 μM	8.0 μM	100 nM	Acetylcholine Calibration Curves:  Dopamine: $K_D = 1.2 \times 10^6$, $n = 23$ Serotonin: $K_D = 1.0 \times 10^5$, $n = 27$ Acetylcholine: $K_D = 0.6 \times 10^6$, $n = 40$ GABA: $K_D = 0.8 \times 10^6$, $n = 32$ Glutamate: $K_D = 1.0 \times 10^6$, $n = 24$ Table 1 (Revised):    Neurotransmitters Dopamine Serotonin Acetylcholine GABA Glutamate     LOD with ss-SERS 1.9 aM 3.0 aM 1.0 aM 99 aM 39 aM   Measured SNR EF (SNR_{ss-SERS}/SNR_{SERS}) @ 1 mM 1.9 × 10³ 1.3 × 10³ 1.5 × 10³ 0.5 × 10³ 0.8 × 10³   Reported LOD with SERS^{27,45} 600 nM 100 nM 4.0 μM 50 μM 600 nM   	Neurotransmitters	Dopamine	Serotonin	Acetylcholine	GABA	Glutamate	LOD with ss-SERS	1.9 aM	3.0 aM	1.0 aM	99 aM	39 aM	Measured SNR EF (SNR _{ss-SERS} /SNR _{SERS}) @ 1 mM	1.9 × 10 ³	1.3 × 10 ³	1.5 × 10 ³	0.5 × 10 ³	0.8 × 10 ³	Reported LOD with SERS ^{27,45}	600 nM	100 nM	4.0 μM	50 μM	600 nM
Neurotransmitters	Dopamine	Serotonin	Acetylcholine	GABA	Glutamate																																													
LOD with ss-SERS	25 aM	1.0 aM	2.1 aM	1.0 aM	10 aM																																													
Measured SNR EF (SNR _{ss-SERS} /SNR _{SERS})	1.6 × 10 ³	2.0 × 10 ³	2.6 × 10 ³	3.0 × 10 ³	2.6 × 10 ³																																													
Reported LOD with SERS	10 nM	10 μM	1.0 μM	8.0 μM	100 nM																																													
Neurotransmitters	Dopamine	Serotonin	Acetylcholine	GABA	Glutamate																																													
LOD with ss-SERS	1.9 aM	3.0 aM	1.0 aM	99 aM	39 aM																																													
Measured SNR EF (SNR _{ss-SERS} /SNR _{SERS}) @ 1 mM	1.9 × 10 ³	1.3 × 10 ³	1.5 × 10 ³	0.5 × 10 ³	0.8 × 10 ³																																													
Reported LOD with SERS ^{27,45}	600 nM	100 nM	4.0 μM	50 μM	600 nM																																													
P.25 Table 1		Table 1. LOD and SNR enhancement of ss-SERS.    Neurotransmitters Dopamine Serotonin Acetylcholine GABA Glutamate      @1402 cm⁻¹ @927 cm⁻¹ @1150 cm⁻¹ @1260 cm⁻¹ @1114 cm⁻¹   LOD with ss-SERS 1.9 aM 3.0 aM 1.0 aM 99 aM 39 aM   Measured SNR EF (SNR_{ss-SERS}/SNR_{SERS}) @ 1 mM 1.9 × 10³ 1.3 × 10³ 1.5 × 10³ 0.5 × 10³ 0.8 × 10³   Reported LOD with SERS^{27,45} 600 nM 100 nM 4.0 μM 50 μM 600 nM   	Neurotransmitters	Dopamine	Serotonin	Acetylcholine	GABA	Glutamate		@1402 cm ⁻¹	@927 cm ⁻¹	@1150 cm ⁻¹	@1260 cm ⁻¹	@1114 cm ⁻¹	LOD with ss-SERS	1.9 aM	3.0 aM	1.0 aM	99 aM	39 aM	Measured SNR EF (SNR _{ss-SERS} /SNR _{SERS}) @ 1 mM	1.9 × 10 ³	1.3 × 10 ³	1.5 × 10 ³	0.5 × 10 ³	0.8 × 10 ³	Reported LOD with SERS ^{27,45}	600 nM	100 nM	4.0 μM	50 μM	600 nM																		
Neurotransmitters	Dopamine	Serotonin	Acetylcholine	GABA	Glutamate																																													
	@1402 cm ⁻¹	@927 cm ⁻¹	@1150 cm ⁻¹	@1260 cm ⁻¹	@1114 cm ⁻¹																																													
LOD with ss-SERS	1.9 aM	3.0 aM	1.0 aM	99 aM	39 aM																																													
Measured SNR EF (SNR _{ss-SERS} /SNR _{SERS}) @ 1 mM	1.9 × 10 ³	1.3 × 10 ³	1.5 × 10 ³	0.5 × 10 ³	0.8 × 10 ³																																													
Reported LOD with SERS ^{27,45}	600 nM	100 nM	4.0 μM	50 μM	600 nM																																													

For the 2nd Reviewer's comments

Detail address:

2.1) Reviewer's comment:

Generally speaking, I am very excited about this article and about its approach; I believe that using these sorts of signal-processing techniques are of critical need for this type of spectroscopy. The results are impressive. I predict that this approach will be utilized extensively across the SERS community, and may trigger additional activity and applications of this type of spectroscopy.

I do believe that the authors need to take care of several issues before this work is ready for acceptance:

There are a number of minor spelling and grammatical mistakes (e.g. on line 15 it should be "... other neurons by releasing..."). I've called out a few here, but there are many places in the text where a grammatical check would be helpful.

Author's response:

As the reviewer pointed out, we strongly believe that this technique can be widely and effectively utilized for highly sensitive and low-cost spectroscopic applications such as Surface-enhanced Raman spectroscopy (SERS), Raman spectroscopy (RS), ultraviolet-visible spectroscopy (UV-VIS), Fourier-transform infrared spectroscopy (FTIR), surface plasmon resonance (SPR), localized surface plasmon resonance (LSPR) or even Terahertz time-domain spectroscopy (THz TDS).

According to the reviewer's comments, the highly qualified native English speaking editors of Springer Nature Author Service (SNAS) edited the manuscript in the proper English language, punctuation, spelling and overall style to improve flow and readability of the text.

2.2) Reviewer’s comment:

a. Line 100: “...(OTDR), except for biomedical sensing applications.” I don’t follow what is meant by “except” here.

Author’s response:

The principles of spread-spectrum technique described in this work have been developed in various ways to increase SNR and improve dynamic range in the field of network technologies such as radar, CDMA, and OTDR. However, the technique has never been applied to any form of biomedical sensing applications, to the best of our knowledge, this work is the first attempt to apply biomedical sensing applications, especially spectroscopy.

In order to avoid the ambiguity of expression, the authors revised the manuscript by removing “except for biomedical sensing applications”.

Page. Column	Original	Author’s Correction
P5.14	This method is inspired from the spread-spectrum techniques, well known for attaining high SNR and dynamic range in network applications such as radio detection and ranging (radar), code-division multiple access (CDMA), and optical time domain reflectometer (OTDR), except for biomedical sensing applications.	This method is based on the spread-spectrum technique, which is well known for attaining a high SNR and dynamic range in network applications such as radio detection and ranging (radar), code-division multiple access (CDMA), or optical time domain reflectometer (OTDR)^{42,43}.

2.3) Reviewer's comment:

In SI Table 5, K_f and n are labelled as Freundlich (note the m instead of n)

Author's response:

Freundlich isotherm is a type of several adsorption isotherm models, studying the relationship which describes adsorption limited to non-ideal and formation of multi-layer. K_F is Freundlich constant and $1/n$ is an empirical parameter related to the adsorption intensity.

The authors revised the supplementary information by correcting spelling errors and unifying symbols in Table 4.

Page. Column	Original	Author's Correction
P29. Supplementary Table 4	$q_e = KC_e^{1/n}$ $\log q_e = \log K_F + \frac{1}{n} \log X_e$ $q_e = \frac{K\chi}{1 + K\chi}$	$q_e = K_F C_e^{1/n}$ $\log q_e = \log K_F + \frac{1}{n} \log C_e$ $q_e = \frac{q_m \cdot K_L \cdot C_e}{1 + K_L \cdot C_e}$

2.4) Reviewer's comment:

It would be useful if the authors could describe in more detail what is meant by (Line 163-164) “The convolution result is transformed into the conventional Raman spectrum of Lorentzian line shape using intensities, center positions, and standard deviations of the decoded SERS peaks...”. In the SI they describe some of the mathematics that goes in to this process, but it is not clear to me if it is a fitting procedure to identify these quantities (intensity, ...) or if they are using previously known literature values (which does not appear to be the case).

Author's response:

Lorentzian Transformation is a fitting procedure, where spectral peaks such as the Raman peaks are fitted to a combination of the Lorentzian line shapes with different widths. The Lorentzian line shape function (Equation on line 14, page 4 of Supplementary Information) calculates the line shape of Raman peak and the line width using the center position of peak, the peak intensity, and the standard deviation of the peak intensity. In this work, all the parameter values are the values obtained from the measured spectra not literature values. That is, after obtaining the peak position (ν_{0i}) and intensity (I_{0i}) values from the decoded spectra (ss-SERS), and repeatedly measuring 5 or more times to calculate the standard deviation (σ_i) of the peak intensity, ss-SERS spectra of the Lorentzian line shape are finally obtained by substituting these parameters into the Lorentzian line shape function.

The table below summarizes parameters used in Lorentzian transform for the ss-SERS spectrum of R6G 5 mM as an example.

Peak assignment	Peak position (ν_{0i})	Intensity (I_{0i})	Standard deviation (σ_i)
C-C ring bending	736	11852	4.0
	775	23727	10.3
	813	1888	6.2
	889	1396	7.0
C-H bending	1077	7804	5.0
	1114	7659	4.7
C-O-C stretching	1223	1553	7.5
	1260	2412	5.1
	1296	721	1.3
C-C stretching	1331	24567	4.0
	1367	1250	1.0
	1403	8599	9.2
	1438	1494	4.7
	1473	11403	3.7
	1508	13835	2.4
	1543	1532	10.4

The authors revised the supplementary information by adding phrases describing the Lorentzian transformation process.

Page. Column	Original	Author's Correction
Supplementary P4.15		Lorentzian Transformation is a fitting procedure, where spectral peaks such as the Raman peaks are fitted to a combination of the Lorentzian line shapes with different widths. After obtaining the peak position (ν_{0i}) and intensity (I_{0i}) values from the decoded spectra (ss-SERS), and repeatedly measuring 5 or more times to calculate the standard deviation (σ_i) of the peak intensity, ss-SERS spectra of the Lorentzian line shape are finally obtained by substituting these parameters into the Lorentzian line shape function (Supplementary Fig. 3).

2.5) Reviewer's comment:

Continuing with the above question, why is this final transform even necessary? Following Step 6 of Sup. Fig. 2, the result is the noise-free SERS signal as a function of position on the spectrometer, which can be directly converted into a measured spectrum. Indeed, this seems to be what is shown in Fig. 2b (top), so why is this final processing step necessary? Is it just to extract the various parameters for each peak?

Author's response:

I fully agree with the reviewer's opinion that the result after Step 6 of Sup. Fig. 2 is the noise-free SERS signal as a function of position on the spectrometer, which can be directly converted into a measured spectrum. However, the low resolution (more than 10 cm^{-1}) of a spectrometer make Raman peaks distorted and broad, thus make it impossible to obtain the exact line shape of the Raman peak.

In general, the substance concentration, the molecular structure, and the crystallinity of material (or the influence of various environment) are extracted from the intensity of a peak, the peak position, and the line width of the peak in a Raman spectrum, respectively. The curve fitting of the Raman spectrum is very important because the line widths of the different Raman peaks vary quite substantially.

The Raman line shape for the molecular vibration of liquid samples fits well in many cases to the Lorentzian profile, which is sharp in the center, but has long wings. Therefore, this work obtains the line widths of peaks and the final line shape by performing Lorentzian curve fitting with parameters such as peak positions, peak intensities, and standard deviation of peak intensities measured from the decoded SERS spectra.

Fig. 2b (top) shows line spectra drawn using spectral data after the transformation of Lorentzian line shape, where the line thickness varies depending on the line width of peaks.

The authors revised the manuscript as follows.

Page. Column	Original	Author's Correction
P8.11	The convolution result is transformed into the conventional Raman spectrum of Lorentzian line shape using intensities, center positions, and standard deviations of the decoded SERS peaks (ss-SERS peaks).	To extract the linewidth of the SERS peaks indicating the crystallinity of the material or the influence of various environments, the ss-SERS spectra of the Lorentzian line shape are finally obtained by substituting the intensities, center positions, and standard deviations of the calibrated decoded SERS peaks into the Lorentzian line shape function (Supplementary Fig. 3).

2.6) Reviewer's comment:

It seems there is more going on as the peak (in the ss-SERS data) at 1368 cm^{-1} in the top panel is highly asymmetric (by eye) while the plotted spectrum shows it to be perfectly symmetric.

Author's response:

The Raman line shape of the isolated molecule has a sharp symmetric profile, but the line shapes of peaks are broadened to different widths due to interaction with various environments like collisions between molecules. The final peak profile is the sum of these individual peak line shapes, and then may be asymmetric.

In the revised ss-SERS spectrum of Figure 2(c), peaks at 775 and 1331 cm^{-1} have asymmetric line shapes.

2.7) Reviewer's comment:

More detail about this process would be useful, and potentially a comparison in the SI of the raw data after processing step 6 with the final output spectrum after the Lorentzian transformation.

Author's response:

The authors explain in detail the process of the Lorentzian transformation by comparing the raw data of Figure 2c (ss-SERS of R6G 5 mM) after processing step 6 with the final spectrum after the Lorentzian transform.

The authors revised the supplementary information by adding a figure describing the Lorentzian transformation process.

Page. Column	Original	Author's Correction
P8.11	The convolution result is transformed into the conventional Raman spectrum of Lorentzian	To extract the linewidth of the SERS peaks indicating the crystallinity of the material or the influence of various environments, the ss-SERS spectra of the Lorentzian line shape are finally

	line shape using intensities, center positions, and standard deviations of the decoded SERS peaks (ss-SERS peaks)	obtained by substituting the intensities, center positions, and standard deviations of the calibrated decoded SERS peaks into the Lorentzian line shape function (Supplementary Fig. 3).
Supplementary P4.16		After obtaining the peak position (ν_{0i}) and intensity (I_{0i}) values from the decoded spectra (ss-SERS), and repeatedly measuring 5 or more times to calculate the standard deviation (σ_i) of the peak intensity, ss-SERS spectra of the Lorentzian line shape are finally obtained by substituting these parameters into the Lorentzian line shape function (Supplementary Fig. 3).

P29
Supplementary
Figure 14

(Author's Correction)

Supplementary Figure 14. Detail description of the Lorentzian line shape transformation in the case of ss-SERS spectrum for R6G 5 mM. The peak width and the final line shape are obtained by performing Lorentzian curve fitting with parameters such as peak positions, peak intensities, and standard deviation of peak intensities measured from the decoded SERS spectra.

2.8) Reviewer's comment:

Finally, to finish this line of questioning, why is it that all the experimental peaks are perfectly Lorentzian in shape? I assume this is related to the aforementioned transform, but it strikes me as very odd that the results would be so perfectly smooth. Even with the impressive approach described in the paper I would still expect some amount of noise to be evident in the Raman spectra, or at least that they not always be so shaped so precisely.

Author's response:

Raman line shape for molecule vibration of liquid samples is known to fit well into Lorentzian profile in many cases³. Curve fitting into Lorentzian line shape makes the line shapes of Raman peaks smooth.

I fully agree the reviewer's concern that some amount of noise is evident in the Raman spectra, or at least that they are not always so shaped so precisely. The ss-SERS method is a technique for restoring the sharp Raman peaks and removing noise by correlation with the spreading code. If the correlation with the spreading code is not perfect due to poor randomness of the code, the correlation error could occur, in which some amount of noise is still included in the spectrum thus the peaks don't become perfectly Lorentzian in the shape. Therefore, it is very important to optimize the spreading code in order to improve performance of noise removal and signal restoration.

2.9) Reviewer's comment:

The discussion on the calculation of the LOD is missing important details – particularly because it is not obvious from Fig. 4b how the plotted curves lead to the reported LOD. For example, serotonin and GABA are claimed to have identical LOD, but the plotted curves have different y-intercepts and by eye similar amounts of noise (both have 2 large error bars, 1 medium, and 1 small), but are listed as having the same LOD. The SI describes the process for finding the LOD but leaves out the values used for the 0 M concentration lines and its associated noise.

Author's response:

The curve of original Figure b includes the ss-SERS signals of the substrate itself, which differs in intensity for each wafer position, so the y-intercept levels and the value of a blank sample (0 M) are different for each neurotransmitter. LOD was calculated by applying different y-intercept levels for each neurotransmitter. As a result, LODs of serotonin and GABA were at the same level.

The authors recalculated LODs of the neurotransmitters with the calibrated ss-SERS spectra that removed the ss-SERS signals of the substrate itself as the final ss-SERS spectra. In the revised Figure 4b, the values of blank samples (0 M) are almost the same, with small error bars at most concentrations of neurotransmitters, and medium error bars at only 1 fM of dopamine and serotonin.

In this revision, for selective detection of primary neurotransmitters, a characteristic Raman peak combination that don't overlap between the neurotransmitters was extracted by using a principal component analysis (PCA). The characteristic Raman peak combination consists of two peaks that have the largest correlation with a specific neurotransmitter in the PCS biplot. The authors confirmed that the characteristic Raman peaks have higher linearity in intensity versus concentration and wider linear dynamic range than those of other peaks.

Neurotransmitters	characteristic Raman peak combination	
	1 st peak (cm ⁻¹)	2 nd peak (cm ⁻¹)
Dopamine	1402	1331
Serotonin	927	1367
Acetylcholine	1150	927
GABA	1114	1260
Glutamate	1114	1150

The LODs of the neurotransmitters at the characteristic Raman peaks were calculated by the method described in Supplementary Information (page 9). The LOD is defined as the concentration corresponding to the intercept between the fit curve of the Freundlich isotherm and the value of a blank sample (0 M) plus three times its standard deviation noise. From the revised Figure 4b of the intensity curve versus concentration, LODs for neurotransmitters were re-calculated and listed in Table 1. In the calibrated ss-SERS spectra where the value of the blank sample (0 M) is removed, the values of a blank sample (0 M) at the characteristic Raman peaks are considered as zero and standard deviation noises (σ) of the peaks at 1402 cm⁻¹, 927 cm⁻¹, 1150 cm⁻¹, 1260 cm⁻¹, and 1114 cm⁻¹ are 16816, 20031, 14646, 18268, and 13258, respectively.

The authors revised the manuscript and supplementary information as follows.

Page. Column	Original	Author's Correction
Manuscript P13.6		A characteristic Raman peak combination for the selective detection of primary neurotransmitters was further extracted by using a PCA (Supplementary Fig. 12). The intensity of characteristic Raman peaks exhibits higher linearity with the sample concentration as well as wider dynamic range than those of other peaks.
Manuscript P13.10	The peak intensities of the ss-SERS for major neurotransmitters: dopamine at 1402 cm ⁻¹ (C-H wagging and N-H twisting), serotonin at 776 cm ⁻¹ (Indole ring stretching), acetylcholine at 776 cm ⁻¹ (N ⁺ C ³ symmetric stretching and C-N stretching), GABA at 779 cm ⁻¹ (COO ⁻ deformation), and glutamate at 1114 cm ⁻¹ (CH ₂ rocking), are clearly three orders of magnitude higher than those of the SERS.	The experimental results indicate that dopamine at 1402 cm ⁻¹ (C-H wagging and N-H twisting), serotonin at 927 cm ⁻¹ (out of phase breathing), acetylcholine at 1150 cm ⁻¹ (CH ₃ rocking and CH ₂ wagging), GABA at 1260 cm ⁻¹ (CH ₂ wagging), and glutamate at 1114 cm ⁻¹ (CH ₂ rocking), are clearly three orders of magnitude higher than those of SERS.
Manuscript P13.17	The acetylcholine molecules are associated with the ss-SERS peaks of 776, 1114, 1331 cm ⁻¹ , and readily detected at 1 aM level, particularly for 776 cm ⁻¹ peak.	The acetylcholine molecules are associated with ss-SERS peaks of 776, 927, 1114, 1150, 1331 cm ⁻¹ and are readily detected at a single attomole level, particularly the 1150 cm ⁻¹ peak.
Manuscript P13.21	GABA of the calming neurotransmitter and serotonin of the mood neurotransmitter in saline solution show the LOD of 1 aM. In addition, acetylcholine of the learning neurotransmitter, glutamate of the memory neurotransmitter, and dopamine of the pleasure neurotransmitter display the LODs of 2 aM, 10 aM, and 25 aM, respectively (See more detail method for the LOD calculation in Supplementary Table 5 and Supplementary Fig. 11).	Acetylcholine of the learning neurotransmitter in saline solution shows an LOD of 1 aM. In addition, GABA of the calming neurotransmitter, serotonin of the mood neurotransmitter, glutamate of the memory neurotransmitter, and dopamine of the pleasure neurotransmitter display LODs of 99 aM, 3 aM, 39 aM, and 1.9 aM, respectively (see more detail method for the LOD calculation in Supplementary Table 4 , Supplementary Fig. 14 , and Supplementary Fig. 15).
P21 Supplementary Figure 12	(Author's Correction)	

Supplementary Figure 12. Extraction of characteristic Raman peak combination for neurotransmitters of dopamine, serotonin, acetylcholine, GABA, and glutamate. The characteristic Raman peak combination consists of two peaks that have the largest correlation with a specific neurotransmitter in the PCS biplot.

P25
Supplementary
Figure 15

Supplementary Figure 15. LOD calculation for the primary neurotransmitters at the characteristic Raman peaks. In the calibrated ss-SERS spectra where the value of the blank sample (0 M) is removed, the values of a blank sample (0 M) at the characteristic Raman peaks are zero and standard deviation noises (σ) of the peaks at 1402 cm⁻¹, 927 cm⁻¹, 1150 cm⁻¹, 1260 cm⁻¹, and 1114 cm⁻¹ are 16816, 20031, 14646, 18268, and 13258, respectively.

- 1 Tuschel, D. Why Are the Raman Spectra of Crystalline and Amorphous Solids Different? *Spectroscopy* **32** 26–33 (2017).
- 2 Manghnani, M. H. *et al.* Raman, Brillouin, and nuclear magnetic resonance spectroscopic studies on shocked borosilicate glass. *Journal of Applied Physics* **109** (2011).
- 3 Bradley, M. Curve fitting in Raman and IR spectroscopy: basic theory of line shapes and applications. **Appl Note 50733** (2018).

Reviewer #1 (Remarks to the Author):

The authors have addressed the concerns from the previous review. I appreciate the effort made to identify the spurious peaks and correcting for the substrate effects. The spectra are so clean now with the distracting peaks removed. One minor edit needs to be made before publication, in Figure S2, the spectra are labelled as "speactrum", that should be corrected. The manuscript is recommended for publication.

Reviewer #2 (Remarks to the Author):

1. In the revised Supplementary Information, the authors describe the fitting process they use to produce the Lorentzian lineshapes from the ss-Raman signals. Could the authors provide a reference for how this process is able to accurately determine the spectral width of a Raman signal using a detector that is limited to $\sim 10 \text{ cm}^{-1}$ as indicated in the response letter?

2. The authors note that their CCD was limited to 250 MHz -- this is quite a high speed for a typical CCD device. In particular, the specification for the spectrometer listed (Hamamatsu C10083CAH) indicates that it has a minimum integration time of 10 ms, longer than the speed needed for a 250 MHz refresh rate. Can the authors explain how they were triggering / reading out from their spectrometer so much faster than its apparent listed speed?

Title: “Spread Spectrum SERS allows label-free detection of attomolar neurotransmitters”

Wonkyoung Lee^{1, 2, 3}, Byoung-Hoon Kang^{1, 2}, Hyunwoo Yang^{1, 2}, Moonseong Park¹, Ji Hyun Kwak¹, Taerin Chung¹, Yong Jeong^{1, 2}, Bong Kyu Kim³, and Ki-Hun Jeong^{1, 2*}

¹Department of Bio and Brain Engineering, Korea Advanced Institute of Science and Technology (KAIST), 291 Daehak-ro, Yuseong-gu, Daejeon 305-701, Republic of Korea

²KAIST Institute for Health Science and Technology (KIHST), KAIST, 291 Daehak-ro, Yuseong-gu, Daejeon 305-701, Republic of Korea

³Bio-Medical IT Convergence Research Division, Electronics and Telecommunications Research Institute (ETRI), 218 Gajeong-ro, Yuseong-gu, Daejeon, 350-700, Republic of Korea

*Corresponding author: kjeong@kaist.ac.kr

Author’s response for the Reviewer’s comments:

We deeply appreciate the editor and reviewers for their interest in our manuscript and their constructive comments. As the reviewer pointed out, we strongly believe that this technique can be widely and effectively utilized for highly sensitive and low-cost spectroscopic applications such as SERS, RS, UV-VIS, FTIR, SPR, LSPR or even THz TDS.

The main concern of one reviewer is how to trigger/read out signals amplitude-modulated at much faster rate than a long integration time/readout time from the spectrometer. i.e., How to trigger and read out amplitude-modulated signals at high modulation rate of 250 MHz in a spectrometer with a minimum CCD integration time of 10 msec?

We handle the amplitude-modulated signals in the spectral domain, not in the time domain, then apply Fourier transform-based convolution theorem to the decoding process for effectively utilizing the advantage of the spread spectrum technique. The frequency components of coded RS/SERS signals are dispersed by a grating and detected by each pixel of CCD of the spectrometer as wavelength components and finally accumulated for a preset integration time. In particular, both CMOS and CCD line cameras can be utilized for this method, however, CCDs operating in the global shutter mode facilitate the implementation of ss-RS/SERS. To synchronize the start time of the CCD exposure with the first bit of the spreading code, the spectrometer triggers using the external trigger signal of the function generator, which sends a voltage signal of the code pattern to the modulator. The stop time of the CCD exposure is determined by the integration time. The output signals of the spectrometer are converted into spatial frequencies by the Fourier transform (in the right side of the below figure). The modulation frequency is limited only by the spectral sampling interval (0.4 nm) and spectral resolution (1 nm) related to the number of CCD pixels (2,048). The integration time (minimum 10 ms) and the readout time (about 5.5 ms) of the CCD do not limit the modulation frequency. The spectral sampling interval of the spectrometer is 0.4 nm, which is a large enough number

of samples to monitor spectral changes due to the modulation frequency of 250 MHz. This detail description has also been added in Supplementary Figure 2 and therefore all the supporting information figures are re-numbered.

Data acquisition from the spectrometer for the Fourier transform-based convolution in the decoding process of the ss-SERS

Including the main concern, we have faithfully addressed the reviewer's point-by-point comment in the attached author response as well as added more detail description in the revised manuscript. In addition, according to the reviewer's comments, we have added a reference explaining the fitting process of the Lorentzian line shape transformation in the reference of the manuscript and corrected a spelling error in Figure S2 of the supplementary information.

For the 1st Reviewer's comments

Detail address:

Reviewer's comment:

The authors have addressed the concerns from the previous review. I appreciate the effort made to identify the spurious peaks and correcting for the substrate effects. The spectra are so clean now with the distracting peaks removed. One minor edit needs to be made before publication, in Figure S2, the spectra are labelled as "spectrum", that should be corrected. The manuscript is recommended for publication.

Author's response:

According to the reviewer's comment, the authors revised the supporting information by correcting "spectrum" into "spectrum" in Figure S2.

The revised supporting information are attached below.

(Author's correction)

For the 2nd Reviewer's comments

Detail address:

2.1) Reviewer's comment:

In the revised Supplementary Information, the authors describe the fitting process they use to produce the Lorentzian lineshapes from the ss-Raman signals. Could the authors provide a reference for how this process is able to accurately determine the spectral width of a Raman signal using a detector that is limited to $\sim 10\text{ cm}^{-1}$ as indicated in the response letter?

Author's response:

A reference explaining the fitting process of the Lorentzian lineshape are as follows.

P.A. Mosier-Boss, S.H. Lieberman, and R. Newbery, "Fluorescence Rejection in Raman Spectroscopy by Shifted-Spectra, Edge Detection, and FFT Filtering Techniques," Applied Spectroscopy, Vol. 49(5), pp. 630-638 (1995)

The reference describes how to accurately fit the lineshapes of the Raman peaks at the top of the broad fluorescence envelope.

We added the reference to the part related to Lorentzian transformation in the manuscript.

Page. Column	Original	Author's Correction
P8.15	To extract the linewidth of the SERS peaks indicating the crystallinity of the material or the influence of various environments, the ss-SERS spectra of the Lorentzian line shape are finally obtained by substituting the intensities, center positions, and standard deviations of the calibrated decoded SERS peaks into the Lorentzian line shape function (Supplementary Fig. 3).	To extract the linewidth of the SERS peaks indicating the crystallinity of the material or the influence of various environments, the ss-SERS spectra of the Lorentzian line shape are finally obtained by substituting the intensities, center positions, and standard deviations of the calibrated decoded SERS peaks into the Lorentzian line shape function (Supplementary Fig. 3) ⁴⁵ .
P16.55 References		45 P. A. Mosier-Boss, S. H. L., R. Newbery. Fluorescence Rejection in Raman Spectroscopy by Shifted-Spectra, Edge Detection, and FFT Filtering Techniques. Appl. Spectrosc. 49, 630-638 (1995).

2.2) Reviewer's comment:

The authors note that their CCD was limited to 250 MHz – this is quite a high speed for a typical CCD device. In particular, the specification for the spectrometer listed (Hamamatsu C10083CAH) indicates that it has a minimum integration time of 10 ms, longer than the speed needed for a 250 MHz refresh rate. Can the authors explain how they were triggering / reading out from their spectrometer so much faster than its apparent listed speed?

Author's response:

The key purpose of using a spectrometer in this work is to handle the amplitude-modulated signals in the spectral domain, not in the time domain, then we apply Fourier transform-based convolution theorem to the decoding process and utilize the advantage of spread spectrum technique, effectively. The frequency components of the modulated SERS signals are dispersed by a grating device and detected by each pixel of CCD of the spectrometer as wavelength components and accumulated for a preset integration time. In particular, both CMOS and CCD line cameras can be utilized for this method, however, CCDs operating in the global shutter mode facilitate in implementation of the ss-SERS. To synchronize the start time of the CCD exposure with the first bit of the spreading code, the spectrometer triggers using the external trigger signal of the function generator, which sends a voltage signal of the code pattern to the modulator. The stop time of the CCD exposure is determined by the integration time. The output signals of the spectrometer are converted into spatial frequencies by the Fourier transform (in the figure below).

The modulation frequency is limited only by the spectral sampling interval and spectral resolution related to the number of CCD pixels. The integration time and the readout time of the CCD do not limit the modulation frequency. The specification for the spectrometer (Hamamatsu C10083CAH) indicates that it has a minimum integration time of 10 ms, a spectral response range from 320 to 1000 nm, 2048 pixels. The spectral sampling interval of the spectrometer is 0.4 nm, which is a large enough number of samples to monitor spectral changes due to the modulation frequency of 250 MHz.

To clarify the triggering / reading process on the spectrometer in response to the reviewer's comment, the authors revised the supporting information and the manuscript as follows.

Pag4e. Column	Author's Correction (Addition)	
P8.9	The decoding process restores quasi noise-free SERS signals by correlating the detected signals with the identical PN code (see the details for the ss-SERS measurement procedure in SI, Supplementary Fig. 1).	The decoding process restores quasi noise-free SERS signals by correlating the detected signals with the identical PN code (see the details for the ss-SERS measurement procedure in SI, Supplementary Fig. 1 and Supplementary Fig. 2).
P4.7 SI	Unlike conventional SERS, the frequency components of the amplitude-modulated SERS signals are dispersed by a grating device and detected by each pixel of CCD of the spectrometer as wavelength components and accumulated for a preset integration time. In particular, CMOS and CCD line cameras can be utilized for this method, however, CCDs operating in the global shutter mode facilitate in implementation of the ss-SERS. To synchronize the start time of the CCD exposure with the first bit of the spreading code, the spectrometer triggers using the external trigger signal of the function generator, which sends a voltage signal of the code pattern to the modulator. The stop time of the CCD exposure is determined by the preset integration time. The output signals of the spectrometer are converted into spatial frequencies by the Fourier transform (Supplementary Fig. 2).	

P.11
Supplementary
Figure 1

P.12
Supplementary
Figure 2

Supplementary Figure 2. Data acquisition from the spectrometer for the Fourier transform-based convolution in the decoding process of the ss-SERS.

Reviewer #2 (Remarks to the Author):

This manuscript is ready to be accepted for publication.